# SorCS1 inhibits amyloid-*β* binding to neurexin and rescues amyloid-*β*–induced synaptic pathology

Alfred Kihoon Lee[1,2], Nayoung Yi[1,3], Husam Khaled[1,3], Benjamin Feller[1,3], Hideto Takahashi[1,2,3,4]

**Amyloid-*β* oligomers (A*β*Os), toxic peptide aggregates found in Alzheimer's disease, cause synapse pathology. A*β*Os interact with neurexins (NRXs), key synaptic organizers, and this interaction dampens normal trafficking and function of NRXs. Axonal trafficking of NRX is in part regulated by its interaction with SorCS1, a protein sorting receptor, but the impact of SorCS1 regulation of NRXs in A*β* pathology was previously unstudied. Here, we show competition between the SorCS1 ectodomain and A*β*Os for *β*-NRX binding and rescue effects of the SorCS1b isoform on A*β*O-induced synaptic pathology. Like A*β*Os, the SorCS1 ectodomain binds to NRX1*β* through the histidine-rich domain of NRX1*β*, and the SorCS1 ectodomain and A*β*Os compete for NRX1*β* binding. In cultured hippocampal neurons, SorCS1b colocalizes with NRX1*β* on the axon surface, and axonal expression of SorCS1b rescues A*β*O-induced impairment of NRX-mediated presynaptic organization and presynaptic vesicle recycling and A*β*O-induced structural defects in excitatory synapses. Thus, our data suggest a role for SorCS1 in the rescue of A*β*O-induced NRX dysfunction and synaptic pathology, providing the basis for a novel potential therapeutic strategy for Alzheimer's disease.**

## Introduction

Alzheimer's disease (AD) is characterized by the accumulation of toxic amyloid-*β* (A*β*) peptides, a major component of senile plaques in the brains of patients with AD (Lambert et al, 1998; Hardy & Selkoe, 2002; Gong et al, 2003; Kayed et al, 2003; Holtzman et al, 2011; Cline et al, 2018). Early pathological features of AD include synaptic dysfunction and synapse loss, and these correlate with cognitive impairments such as memory loss (Selkoe, 2002; Scheff & Price, 2003; Sheng et al, 2012; Cline et al, 2018). The role of A*β* peptides, especially A*β* oligomers (A*β*Os), in synaptic dysfunction and synapse loss is well studied both in vitro and in vivo. A*β* treatment of cultured hippocampal neurons decreases the levels of pre-/postsynaptic proteins, synaptic vesicle recycling, the density of dendritic spines, and the postsynaptic reception sites of excitatory synapses (Lacor et al, 2004; Roselli et al, 2005; Snyder et al, 2005; Calabrese et al, 2007; Kelly & Ferreira, 2007; Lacor et al, 2007; Nimmrich et al, 2008; Roselli et al, 2009; Russell et al, 2012; Park et al, 2013; Ripoli et al, 2013; He et al, 2019). A*β* treatment of mouse and rat hippocampal slices alters synaptic plasticity, blocking long-term potentiation and enhancing long-term depression (Lambert et al, 1998; Walsh et al, 2002; Wang et al, 2002; Shankar et al, 2007; Shankar et al, 2008; Li et al, 2009; Wei et al, 2010; Mucke & Selkoe, 2012; Sheng et al, 2012). In vivo studies using AD model mice have shown that A*β* overproduction decreases the synaptic protein expression and dendritic spine density and impairs long-term potentiation (Oakley et al, 2006; D'Amelio et al, 2011; Pozueta et al, 2013; Herzer et al, 2018; Suzuki et al, 2020). Thus, synapses are vulnerable to A*β*, and understanding the molecular mechanisms that underlie this vulnerability is crucial for explaining how A*β* induces synapse pathology and how A*β*-induced synapse pathology could be ameliorated.

Synapse formation, maturation, maintenance, and synaptic plasticity all depend on the proper functioning of synaptic-organizing complexes, trans-synaptic adhesion complexes with the ability to promote pre- and/or postsynaptic assembly (Craig & Kang, 2007; Siddiqui & Craig, 2011; Takahashi & Craig, 2013; Bemben et al, 2015; Sudhof, 2021). The neurexin (NRX)–neuroligin (NLGN) complex is the most well-studied synaptic organizing complex (Sudhof, 2008, 2017; Gomez et al, 2021). In a previous study, we demonstrated that, of the known synaptic organizers, only NRX family members bind to A*β*Os (Naito et al, 2017). Specifically, A*β*Os bind to the *β*-isoforms of NRXs (*β*-NRXs) through their N-terminal *β*-NRX-specific histidine-rich domain (HRD) and to *α*- and *β*-NRX1/2 (NRX1*α*/*β*, 2*α*/*β*) that possess the splicing site 4 (S4) insert (Naito et al, 2017). A*β*O binding to the NRX1*β* HRD reduces the surface expression of NRX1*β* on axons, and A*β*O treatment diminishes NRX-mediated excitatory presynaptic differentiation (Naito et al, 2017). These findings suggest A*β*O disruption of NRX trafficking and function as a mechanism underlying AD synaptic pathology.

Intracellular trafficking of NRXs is regulated in part by SorCS1 (Savas et al, 2015; Ribeiro et al, 2019), a member of the vacuolar protein sorting 10 (VPS10)-related sortilin family predominantly expressed in the brain (Hermey et al, 1999; Hermey et al, 2004).

[1]Synapse Development and Plasticity Research Unit, Institut de Recherches Cliniques de Montréal, Montreal, Canada   [2]Integrated Program in Neuroscience, McGill University, Montreal, Canada   [3]Department of Medicine, Université de Montréal, Montreal, Canada   [4]Division of Experimental Medicine, McGill University, Montreal, Canada

Correspondence: Hideto.Takahashi@ircm.qc.ca

Previously isolated in proteomics studies as an NRX1β binding protein (Savas et al, 2015; Traunmuller et al, 2016), SorCS1 interacts with NRX1β through the SorCS1 VPS10 domain to promote the surface expression of NRX1β on axons (Savas et al, 2015). In the APP/PS1 AD mouse model, which displays Aβ overproduction and Aβ plaque formation (Radde et al, 2006), SorCS1 expression decreased in the frontal cerebral cortex and hippocampus (Hermey et al, 2019). Previous genetic studies have also identified a variation of the *SORCS1* gene as a potential risk factor for AD through its effect on the Aβ pathway (Liang et al, 2009; Reitz et al, 2011). Given that SorCS1 interacts with and regulates NRXs, these data suggest that SorCS1 could be involved in Aβ-induced synaptic pathology through regulating NRXs.

In this study, we investigated whether and how SorCS1 regulates NRXs under Aβ pathological conditions and how this could be involved in Aβ-synaptic pathology. First, we characterized the interactions between the SorCS1 ectodomain and NRXs in the presence and absence of AβOs. We defined a domain of NRXs that is responsible for SorCS1 binding and discovered that the SorCS1 ectodomain and AβOs compete for NRX1β binding. Furthermore, an expression of the SorCS1b isoform in axons normalizes the AβO-induced impairment of NRX-mediated presynaptic organization and synaptic vesicle recycling and rescues the AβO-induced structural defects in excitatory synapses. Together, our results suggest that SorCS1 may compete against AβOs for β-NRX binding to alleviate AβO-induced synaptic pathology.

# Results

## The SorCS1 ectodomain interacts with β-NRXs through their N-terminal HRD

To test whether SorCS1 interacts with not only NRXs but also other synaptic organizers, we performed cell surface–binding assays in which a soluble recombinant SorCS1 ectodomain tagged with human immunoglobulin Fc region (SorCS1-Fc) was applied to COS-7 cells expressing one of the synaptic organizers (Fig 1A). We first confirmed that SorCS1-Fc interacted with NRX1β, as previously reported (Savas et al, 2015), but not with CD4, a negative control. We tested a total of 18 synaptic organizers outside of the NRX family and found no bound SorCS1-Fc signal on COS-7 cells expressing any of the other synaptic organizers. These data indicate that, of the known synaptic organizers, NRXs are the only interaction partners of SorCS1. Given the numerous different isoforms of the NRX family, including α/β-isoforms and isoforms with or without an S4 insert (Craig & Kang Y, 2007; Sudhof, 2017; Gomez et al, 2021), we next determined which NRX isoforms interact with SorCS1 (Fig 1B and C). SorCS1-Fc interacted with NRX1β and 2β regardless of S4 insertion but not with NRX1α, 2α, 3α, or NRX3β. We also investigated the binding of SorCS2-Fc proteins to NRX isoforms and found that SorCS2-Fc interacted with NRX1β strongly and interacted with NRX2β and 3β weakly, regardless of S4 insertion, but not with any α-isoforms of NRXs (Fig S1). These data suggest that the interaction of SorCS1/2 with β-NRXs relies on β–NRX-specific domains. Because the HRD is the domain that distinguishes β-NRXs from α-NRXs (Reissner et al, 2013), we next tested whether SorCS1-Fc interacts with NRX1β and 2β lacking their HRD (NRX1βΔHRD and NRX2βΔHRD,

respectively). We found that SorCS1-Fc did not interact with NRX1βΔHRD or NRX2βΔHRD, indicating that the HRDs of NRX1β and 2β are responsible for the SorCS1 interaction (Fig 1B and C). Similarly, SorCS2-Fc did not interact with NRX1βΔHRD, NRX2βΔHRD, or NRX3βΔHRD, indicating the involvement of their HRD in SorCS2–β-NRX interaction (Fig S1). Next, we performed pull-down assays to test the biochemical interaction of SorCS1 with NRX1β through its HRD using purified recombinant SorCS1 ectodomain tagged with 6×His (SorCS1-His) incubated with the NRX1β ectodomain tagged with Fc (NRX1β-Fc), the NRX1β ectodomain lacking its HRD tagged with Fc (NRX1βΔHRD-Fc) or Fc protein as a negative control (Figs 1D and S2). SorCS1-His was co-precipitated with NRX1β-Fc but not with NRX1βΔHRD-Fc or Fc (Fig 1D). Further validation confirmed that the NRX1βΔHRD-Fc proteins are the desired material lacking the HRD (Fig S3) despite their unexpected slower migration in SDS–PAGE (Fc blot in Fig 1D), which is presumably a "gel shifting" phenomenon (Rath et al, 2009). These results provide further support for a protein interaction between the SorCS1 ectodomain and the NRX1β ectodomain through their HRD, consistent with the results of the cell surface–binding assays.

## SorCS1 and AβOs bind competitively to NRX1β

We have previously discovered that the β-NRX HRD is also responsible for the interaction of β-NRX with AβOs but not Aβ monomers (Naito et al, 2017), suggesting the possibility that SorCS1 and AβOs compete for binding to β-NRXs since they share a binding domain on β-NRXs. To test this, we performed cell-based competitive protein binding assays using oligomerized samples of biotin-conjugated Aβ peptides (biotin–AβO) (Fig S4). First, we tested whether and how SorCS1 affects the interaction of AβOs with NRX1β using COS-7 cells expressing NRX1β incubated with a single concentration (100 nM monomer equivalent) of biotin–AβOs in the presence of varying concentrations of SorCS1-Fc (from 0 nM to 1,500 nM). SorCS1-Fc inhibited the binding of biotin-AβOs onto COS-7 cells expressing NRX1β in a dose-dependent manner, whereas Fc, a negative control, had no significant effect on AβO binding to NRX1β (Fig 2A and B). The inhibition curve revealed that the half-maximal inhibition concentration ($IC_{50}$) value for SorCS1 was 92.8 nM (Fig 2B). Conversely, we next tested whether and how AβOs affect the binding of SorCS1 to NRX1β using NRX1β-expressing COS-7 cells incubated with a single concentration (250 nM) of SorCS1-Fc in the presence of varying concentrations of biotin–AβOs (from 0–2,000 nM, monomer equivalent, which corresponds to 0–66 nM 150 kD oligomers). AβOs up to 1,000 nM monomer equivalent had no significant effect on the binding of SorCS1 to NRX1β. However, at 2,000 nM monomer equivalent, AβOs reduced the SorCS1–NRX1β interaction by half (Fig 2C and D). Together, these findings suggest that the SorCS1 ectodomain competes with AβOs for binding to NRX1β.

Given a previous study showing a preferential cis-interaction of SorCS1 and NRX (Savas et al, 2015), we next tested whether a cis-interaction between SorCS1 and NRX1β inhibits AβO–NRX1β binding in cell surface–binding assays using COS-7 cells co-expressing SorCS1b-myc and HA-NRX1β exposed to 250 nM biotin-AβOs (Fig 2E and F). There are several SorCS1 isoforms that share extracellular and transmembrane regions but possess different cytoplasmic tails

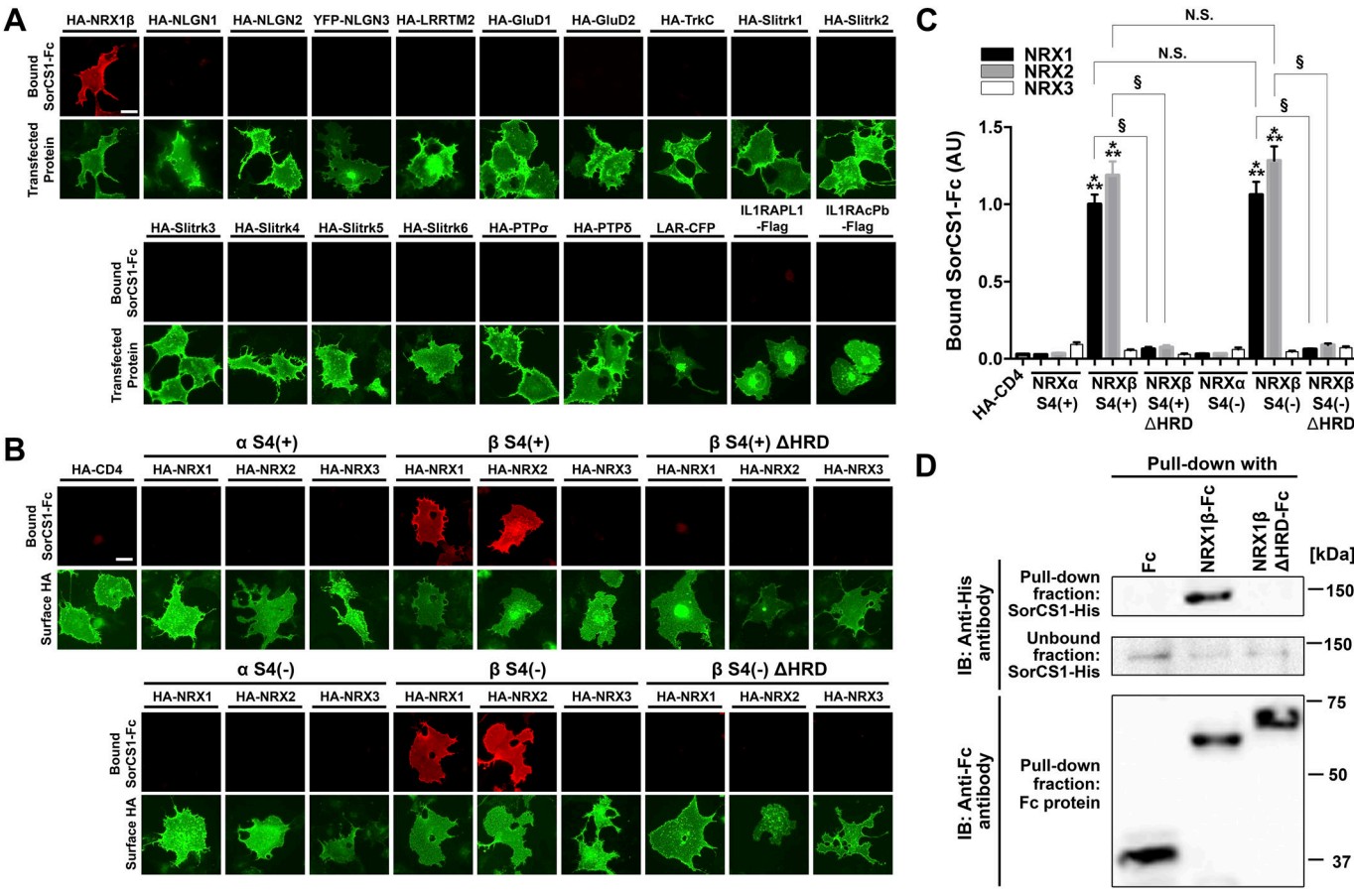

**Figure 1. The SorCS1 ectodomain binds to NRX1β and 2β, depending on their N-terminal histidine-rich domain.**
**(A)** Representative images showing the results of cell surface–binding assay testing for interaction between SorCS1-Fc and known synaptic organizers. SorCS1-Fc (1 µM) was added to COS-7 cells expressing the indicated construct. Note that SorCS1-Fc binds to COS-7 cells expressing HA-NRX1βS4(−), but not to those expressing any of the other organizers. For the N-terminal extracellular HA-tagged constructs, surface HA was immunostained to verify the expression of the construct on the COS-7 cell surface. Scale bars: 30 µm. **(B)** Representative images showing the binding of SorCS1-Fc (1 µM) to COS-7 cells expressing the indicated isoform of extracellularly HA-tagged NRX constructs. S4(+) and S4(−) indicate with and without an insert at splicing site 4, respectively, and ΔHRD indicates lack of the N-terminal histidine-rich domain (HRD) of β-NRX. HA fluorescent signals correspond to surface HA. Scale bar: 30 µm. **(C)** Quantification of bound SorCS1-Fc for each NRX construct. n = 30 cells for each construct from three independent experiments, one-way ANOVA, P < 0.0001. ***P < 0.001 compared with HA-CD4 and §P < 0.001 between NRXβ and NRXβΔHRD by Tukey's multiple comparisons test. Data are presented as mean ± SEM. **(D)** Pull-down assays of purified recombinant His-tagged SorCS1 ectodomain protein with Fc, NRX1βS4(−)-Fc, or NRX1βS4(−)ΔHRD-Fc proteins indicate that the SorCS1 ectodomain and the NRX1β ectodomain form a complex when the NRX1β HRD is present.

(Hermey, 2009). In the present study, we used SorCS1b because this isoform is preferentially expressed on the cell surface in contrast to other SorCS1 isoforms including SorCS1cβ (Hermey et al, 2003; Hermey et al, 2015), which are mainly expressed in endosomal compartments with minimal cell surface expression (Savas et al, 2015; Ribeiro et al, 2019). First, we confirmed that SorCS1b-myc can be expressed on the COS-7 cell surface (Fig S5) and that COS-7 cells transfected with SorCS1b-myc alone showed no apparent AβO binding on their cell surface (Fig S6). These results suggest that when COS-7 cells co-express HA-NRX1β and SorCS1b-myc, bound AβO signals on cell surface correspond to surface HA-NRX1β expression. Next, we found that the AβO-binding signal on COS-7 cells co-expressing SorCS1b-myc and HA-NRX1β was significantly lower than that on COS-7 cells expressing only HA-NRX1β. We also performed AβO-binding assays using COS-7 cells co-transfected with HA-NRX1β and SorCS1b-myc lacking a VPS10 domain, which does not bind to NRX1β (Savas et al, 2015)

(SorCS1bΔVPS10-myc; Fig 2E and F). We found that the co-expression of SorCS1bΔVPS10-myc, which we confirmed was also at the COS-7 cell surface (Fig S5), had no effect on AβO–NRX1β binding (Fig 2E and F). These data support the idea that SorCS1–NRX1β cis-interaction is involved in AβO–NRX1β binding competition. On the other hand, neither the co-expression of SorCS1b-myc nor that of SorCS1bΔVPS10-myc significantly affected the interaction between NLGN1 and NRX1β (Fig 2G and H), suggesting that the SorCS1b–NRX1β cis-interaction had no effect on NRX1β–NLGN1 interaction.

Given that axonal NRXs make trans-synaptic complexes with dendritic NLGN1 (Sudhof, 2008, 2017; Gomez et al, 2021), we next tested whether the SorCS1–NRX1β cis-complex on the axon surface could interact with NLGN1 in trans. To do so, we performed protein-clustering assays using inert beads coated with NLGN1-Fc in primary cultured hippocampal neurons co-transfected to express extracellular HA-tagged NRX1β or HA-NRX1βΔHRD and untagged

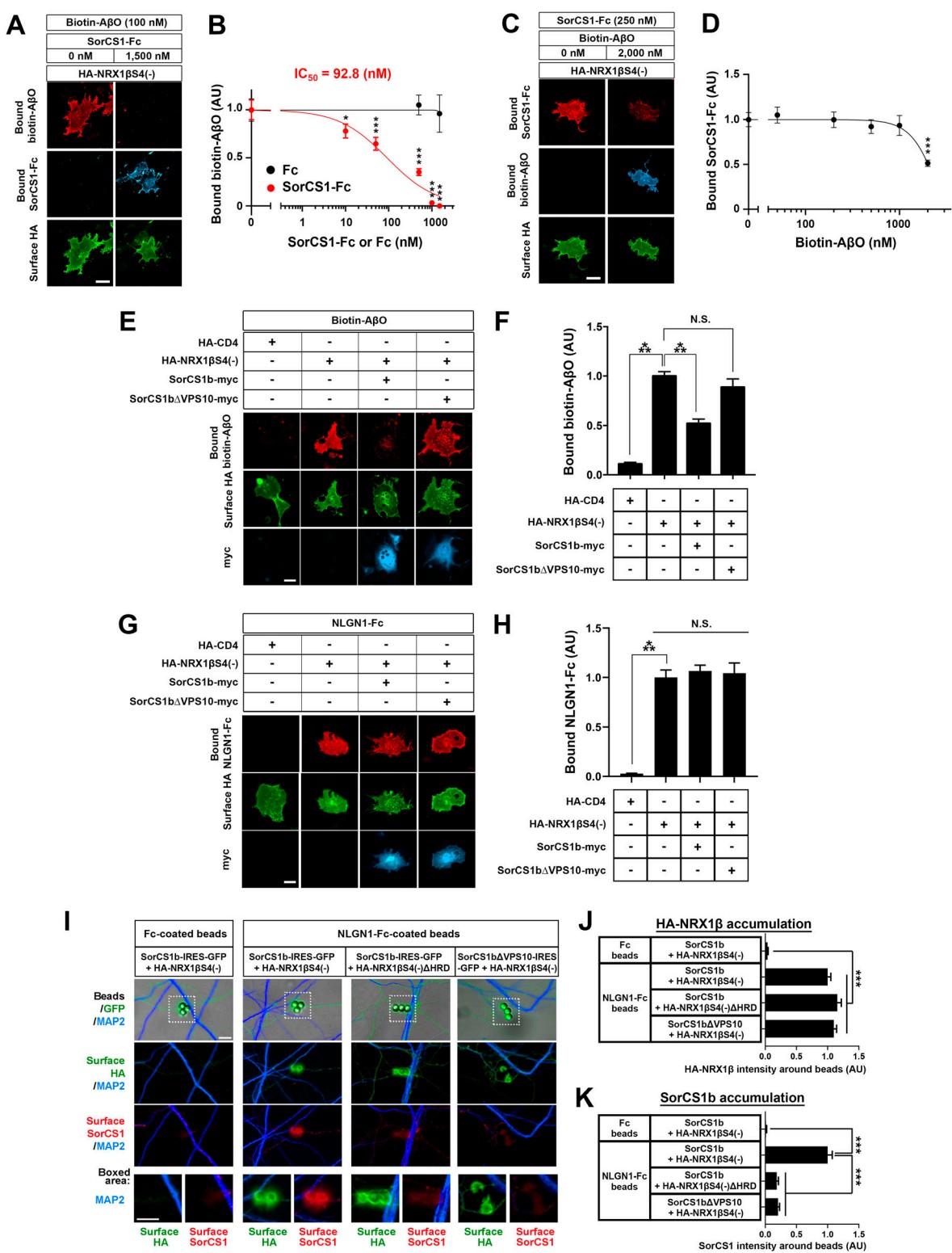

**Figure 2. The SorCS1 ectodomain and AβOs compete for binding to NRX1β.**
**(A)** Representative images of triple-labelling for bound biotin-AβOs, bound SorCS1-Fc, and surface HA of COS-7 cells expressing extracellularly HA-tagged NRX1βS4(−). Recombinant biotin-AβOs (100 nM, monomer equivalent) with/without SorCS1-Fc (1,500 nM) were extracellularly applied to COS-7 cells expressing HA-NRX1βS4(−). **(B)** Quantification of biotin-AβOs bound to COS-7 cells expressing HA-NRX1βS4(−) in the presence of various concentrations of SorCS1-Fc (0–1,500 nM). The half maximal inhibitory concentration ($IC_{50}$) is 92.8 nM. **(C)** Representative images of triple-labelling for bound SorCS1-Fc, bound biotin-AβOs, and surface HA of COS-7 cells expressing HA-NRX1βS4(−). SorCS1-Fc (250 nM) with/without biotin-AβOs (2,000 nM, monomer equivalent) was applied to COS-7 cells expressing HA-NRX1βS4(−). **(D)** Quantification of

SorCS1b or SorCS1bΔVPS10 together with GFP (Fig 2I–K). By surface immunostaining for SorCS1b and HA-NRX1β without cell permeablization, we found that NLGN1-coated beads recruit the co-accumulation of HA-NRX1β and SorCS1b on the surface of contacting axons, but this recruitment fails in the presence of HA-NRX1βΔHRD or SorCS1bΔVPS10 (Fig 2I–K). Given that SorCS1 does not bind to NLGN1 (Fig 1A), these data suggest that SorCS1 associates with the NRX1β–NLGN1 trans-complex via SorCS1–NRX1β cis-interaction. Altogether, these findings suggest that the cis-interaction of SorCS1 with NRX1β may compete with AβOs for binding to NRX1β without affecting NLGN1–NRX1β binding, which would be beneficial for rescuing AβO-induced dysfunction of NRX-based synaptic organizer complexes.

### SorCS1b is expressed on the axon surface and colocalizes with NRX1β through their extracellular interaction

Given that NRXs carry out their functions in synapse organization predominantly by being expressed on the axon surface (Dean et al, 2003; Sudhof, 2008, 2017; Gomez et al, 2021), we tested whether SorCS1b is also expressed on the axon surface by surface immunostaining of the SorCS1 extracellular domain in primary-cultured hippocampal neurons transfected with SorCS1b-IRES-GFP. Our immunocytochemical data show significant expression of SorCS1b at the axon surface as well as the dendrite surface (Fig 3A and B). As a negative control, there was no apparent surface SorCS1 signal in neurons transfected with the empty IRES-GFP vector (Fig 3A and B), indicating that the surface SorCS1 signals observed in neurons transfected with SorCS1b-IRES-GFP are due to exogenous SorCS1b expression rather than non-specific signals or endogenous SorCS1 expression. Next, we tested if SorCS1b colocalized with NRX1β using neurons co-transfected with SorCS1b-IRES-GFP or SorCS1bΔVPS10-IRES-GFP and a plasmid expressing extracellular HA-tagged NRX1β or HA-NRX1βΔHRD. Using Pearson's correlation coefficient (r) to measure the degree of correspondence between images of the two signals showed that surface SorCS1b highly colocalized with surface HA-NRX1β on axons (r = 0.84 ± 0.01), especially at the contact sites between axons and dendrites (arrowheads in Fig 3C and D). The colocalization was significantly reduced when HA-NRX1β lacking the HRD (r = 0.53 ± 0.02) or SorCS1b lacking the VPS10 domain (r = 0.52 ± 0.03) were present (Fig 3C and D). Given that the NRX1β HRD and SorCS1bVPS10 are responsible for extracellular SorCS1–NRX1β interaction (Fig 1; Savas et al, 2015), these data suggest that SorCS1b colocalizes with NRX1β on the axon surface through their

extracellular interaction. Together with our findings in binding assays on non-neuronal cells (Figs 1 and 2), our results in co-transfected neurons also support the cis-interaction of SorCS1b with NRX1β on the axon surface.

### SorCS1b expression in axons rescues AβO-induced impairment of NRX-mediated presynaptic differentiation

NRXs mediate presynaptic differentiation induced by NLGNs and LRRTM1/2/3 (Sudhof, 2017; Gomez et al, 2021), and AβO treatment diminishes NRX-mediated presynaptic differentiation by reducing the surface expression of β-NRXs on axons (Naito et al, 2017). Therefore, we next investigated whether and how exogenous SorCS1b expression in axons affects AβO-induced impairment of NRX-mediated presynaptic differentiation in artificial synapse-formation assays using Fc protein-coated inert beads (Figs 4 and S7). Beads coated with NLGN1-Fc, LRRTM2-Fc, or Fc (a negative control) were applied together with treatments of either AβOs (500 nM monomer equivalent) or vehicle control for 24 h to cultured hippocampal neurons transfected with vectors co-expressing SorCS1b and GFP (SorCS1-IRES-GFP), SorCS1bΔVPS10, and GFP (SorCS1bΔVPS10-IRES-GFP), or GFP alone as a control (IRES-GFP) (Fig 4A–D). Subsequently, VGLUT1 accumulation at contact sites between the beads and GFP-positive axons was measured to assess the presynaptic induction activity of NLGN1 and LRRTM2 (Fig 4A–D). As previously reported (de Wit et al, 2009; Ko et al, 2009; Naito et al, 2017), in the absence of AβOs (vehicle treatment), beads coated with NLGN1-Fc or LRRTM2-Fc induced strong accumulation of VGLUT1 in contacting axons expressing only GFP (Fig 4A–D). Furthermore, as we previously reported (Naito et al, 2017), AβO treatment significantly decreased VGLUT1 accumulation induced by NLGN1 and LRRTM2 in contacting axons expressing only GFP (Fig 4A–D). Remarkably, AβO treatment failed to decrease VGLUT1 accumulation induced by NLGN1 and LRRTM2 in contacting axons expressing SorCS1b, suggesting a rescue effect of SorCS1 on AβO-induced impairment of NLGN1 and LRRTM2 presynaptic induction activity (Fig 4A–D). This effect was not detected in contacting axons expressing the β-NRX binding–dead SorCS1bΔVPS10 variant, suggesting that the rescue effect of SorCS1b relies on SorCS1-β-NRX extracellular interaction (Fig 4A–D). Indeed, high colocalization of SorCS1b with HA-NRX1β on the axon surface was maintained even after AβO treatment (500 nM monomer equivalent for 24 h) (Fig S8). We also confirmed that the targeting of SorCS1bΔVPS10 to the axon surface was equivalent to that of SorCS1b, suggesting that the

---

SorCS1-Fc bound to COS-7 cells expressing HA-NRX1βS4(−) in the presence of various concentrations of biotin-AβOs (0–2,000 nM, monomer equivalent). **(E)** Representative images of triple-labelling for bound biotin-AβOs, surface HA, and total myc (surface and intracellular myc both) of COS-7 cells co-expressing HA-NRX1βS4(−) with either intracellularly myc-tagged-SorCS1b (SorCS1b-myc) or SorCS1b-myc lacking the VPS10 domain (SorCS1bΔVPS10-myc). Biotin-AβOs (250 nM, monomer equivalent) were applied to COS-7 cells co-expressing HA-NRX1βS4(−) with SorCS1b-myc or SorCS1bΔVPS10-myc. COS-7 cells expressing HA-CD4 were used as a negative control. **(F)** Quantification of biotin-AβOs bound to COS-7 cells co-expressing HA-NRX1βS4(−) with SorCS1b-myc or SorCS1bΔVPS10-myc. **(G)** Representative images of triple-labelling for bound NLGN1-Fc, surface HA, and total myc of COS-7 cells co-expressing HA-NRX1βS4(−) with SorCS1b-myc or SorCS1bΔVPS10-myc. NLGN1-Fc (20 nM) was applied to COS-7 cells expressing the indicated constructs. **(H)** Quantification of NLGN1-Fc bound to COS-7 cells co-expressing HA-NRX1βS4(−) with SorCS1b-myc or SorCS1bΔVPS10-myc. **(I)** Representative images obtained in protein-clustering assays using neuroligin 1 (NLGN1)-Fc–coated beads or Fc-coated beads (a negative control) in cultures of hippocampal neurons (DIV21) co-transfected with either SorCS1b-IRES-GFP and HA-NRX1βS4(−), SorCS1b-IRES-GFP and HA-NRX1βS4(−)ΔHRD, or SorCS1bΔVPS10-IRES-GFP and HA-NRX1βS4(−). **(J, K)** Quantification of the average intensity of HA-NRX1β or HA-NRX1βΔHRD (J) and SorCS1b or SorCS1bΔVPS10 (K) around beads coated with NLGN1-Fc or Fc. n = 30 cells for each condition from three independent experiments for (B, D, F, H) and n > 100 beads for each condition from three independent experiments for (J, K), one-way ANOVA, P < 0.0001. ***P < 0.001, *P < 0.05, N.S., not significant by Dunnett's test compared with the 0 nM control condition for (B, D) and Tukey's multiple comparisons test for (F, H, J, K). Data are presented as mean ± SEM. Scale bar: 30 μm (A, C, E, G) and 5 μm (I).

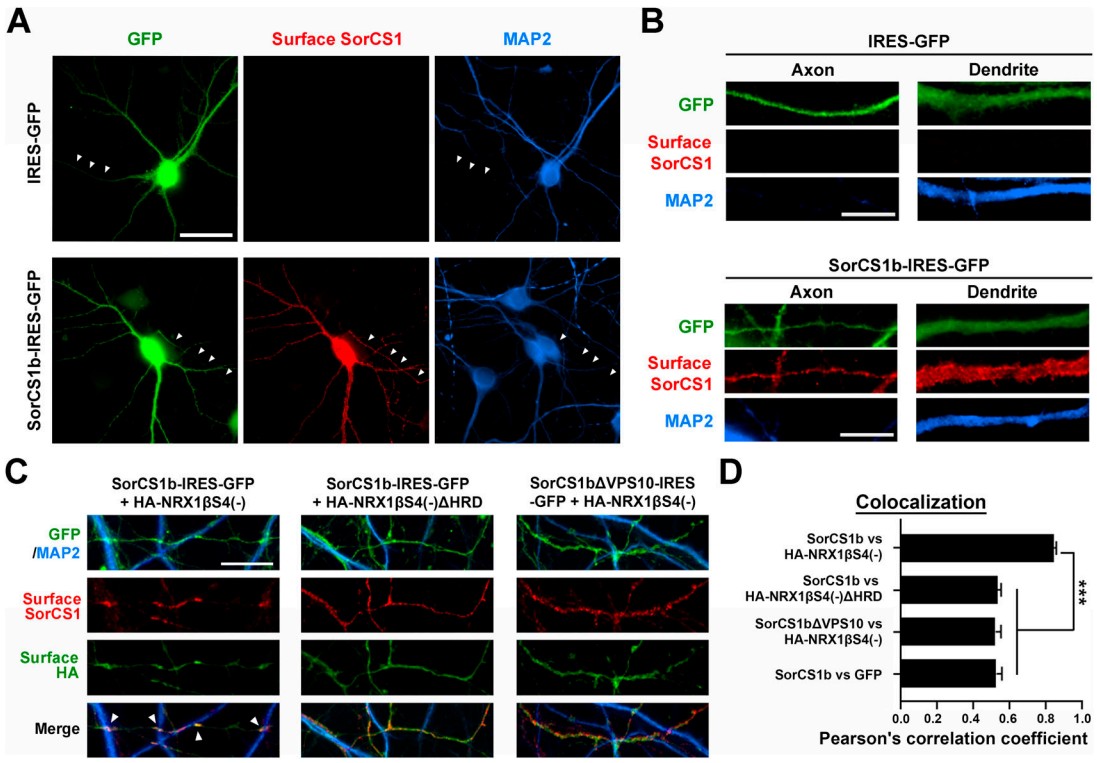

**Figure 3. SorCS1b is targeted to the axon surface of cultured hippocampal neurons where it colocalizes with NRX1β.**
**(A, B)** Representative images of cultured hippocampal neurons (DIV21) transfected with the IRES-GFP or SorCS1b-IRES-GFP expression vectors followed by immunostaining of surface SorCS1 and MAP2 before and after cell permeabilization, respectively. The GFP and MAP2 signals were used to distinguish axons (GFP-positive but MAP-negative neurites, arrowheads in (A)) from dendrites (GFP- and MAP2-positive neurites). Immunoreactivity for surface SorCS1 was detected in both axons and dendrites of neurons transfected with SorCS1b-IRES-GFP, but not in those transfected with IRES-GFP. **(C)** Representative images showing the axons of cultured hippocampal neurons (DIV21) co-transfected with SorCS1b-IRES-GFP and HA-NRX1βS4(−) (left), SorCS1b-IRES-GFP and HA-NRX1βS4(−)ΔHRD (middle), or SorCS1bΔVPS10-IRES-GFP and HA-NRX1βS4(−) (right) and immunostained for surface SorCS1 and surface HA before permeabilization and MAP2 after permeabilization. SorCS1b is nicely colocalized with HA-NRX1β (left), but not HA-NRX1βΔHRD (middle), especially at the contact sites between GFP-expressing axons and dendrites (MAP2-positive neurites) (arrows). **(D)** Quantification of colocalization between the indicated proteins using Pearson's correlation coefficients. n = 30 cells for each condition from three independent experiments, one-way ANOVA, $P < 0.0001$, and ***$P < 0.001$ by Tukey's multiple comparisons test. Scale bar: 30 μm (A) and 10 μm (B, C).

absence of a rescue effect in the presence of SorCS1bΔVPS10 is not due to insufficient expression on the axon surface (Fig S9). We next investigated the effects of SorCS1b on another class of synaptic-organizing complex in which type IIa receptor-type protein tyrosine phosphatases (RPTPs) such as PTPσ, PTPδ, and LAR mediate the presynaptic induction activity of Slitrk1-6 and TrkC (Takahashi & Craig, 2013). We performed the same artificial synapse formation assays using Slitrk2-Fc–coated beads to check for the effects of SorCS1b on RPTP-mediated presynaptic differentiation. We found that axonal expression of either SorCS1b or SorCS1bΔVPS10 did not significantly affect Slitrk2-induced VGLUT1 accumulation regardless of AβO treatment (Fig 4E and F). This finding is in line with the results of our previous study showing that RPTP-mediated presynaptic differentiation is insensitive to AβOs (Naito et al, 2017; Lee et al, 2020) and suggests that it is also insensitive to SorCS1, consistent with our binding results showing that SorCS1-Fc does not interact with any RPTPs or Slitrks (Fig 1A). In conclusion, these findings suggest that axonal SorCS1b expression rescues AβO-induced impairment of NRX-mediated presynaptic organization through cis-interaction with β-NRXs.

## SorCS1b expression in axons rescues AβO-mediated impaired presynaptic vesicle recycling

According to previous studies in hippocampal neurons, β-NRXs positively regulate the neurotransmitter release at excitatory synapses (Anderson et al, 2015), but AβOs suppress the excitatory neurotransmitter release (Nimmrich et al, 2008; Parodi et al, 2010; He et al, 2019) and disrupt synaptic vesicle endocytosis (Kelly & Ferreira, 2007; Park et al, 2013). Given the competition between the SorCS1 ectodomain and AβOs for NRX1β binding (Fig 2) and the rescue effects of SorCS1b on AβO-induced impaired function of NRXs (Fig 4), we next tested whether exogenous SorCS1b expression in axons rescues AβO-induced impaired synaptic vesicle endocytosis through β-NRX interaction in live transfected hippocampal neurons expressing SorCS1b and GFP, SorCS1bΔVPS10 and GFP, or GFP alone (Fig 5). To do so, we assessed the uptake of an antibody directed against the synaptotagmin-1 luminal domain (Syn-Tag1), which occurs only during active recycling of synaptic vesicles (Malgaroli et al, 1995; Ammendrup-Johnsen et al, 2015), in transfected (GFP-positive) axons innervating non-transfected (GFP-negative) dendrites to investigate the effect of axonal, but not

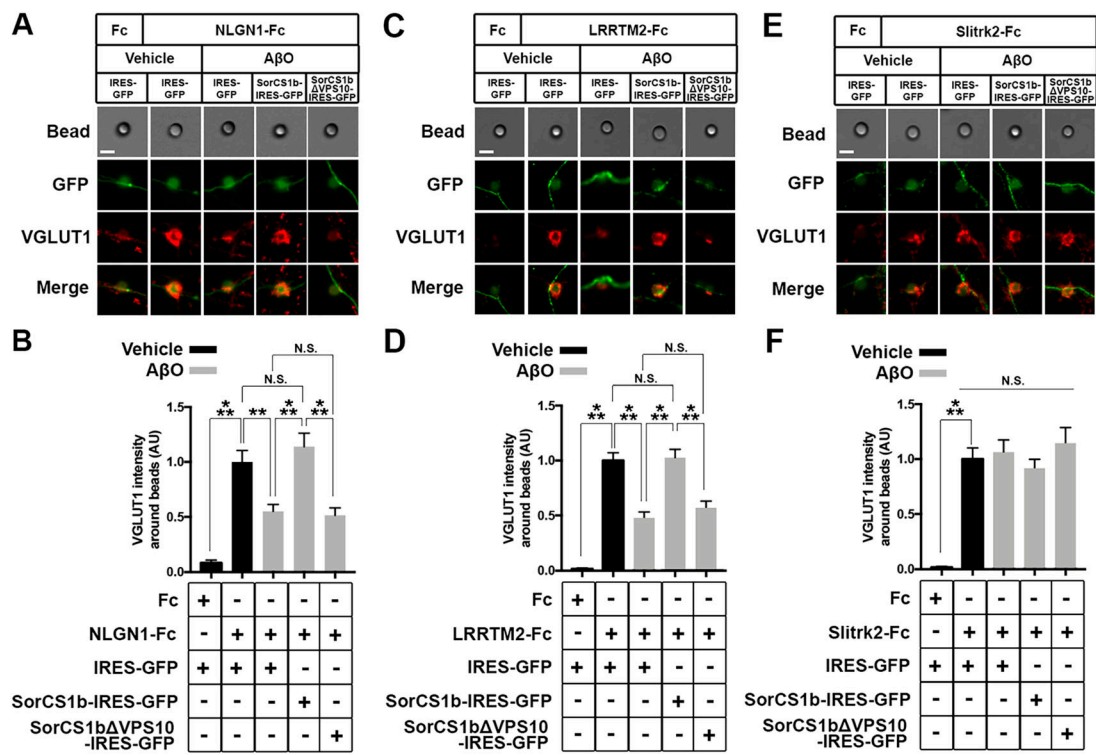

**Figure 4. Axonal expression of SorCS1b restores AβO-induced impairment of NRX-mediated excitatory presynaptic differentiation.**
**(A, C, E)** Representative images of artificial synapse formation assays using cultured hippocampal neurons transfected with IRES-GFP, SorCS1b-IRES-GFP, or SorCS1bΔVPS10-IRES-GFP. The neurons were treated with inert protein-G beads coated with NLGN1-Fc (A), LRRTM2-Fc (C), Slitrk2-Fc (E), or Fc; a negative control protein (left in A, C, E) together with AβOs (500 nM, monomer equivalent) or vehicle. 24 h after the treatment, the neurons were immunostained for the excitatory presynaptic marker VGLUT1. Scale bars: 5 µm. **(B, D, F)** Quantification of VGLUT1 intensity around the beads coated with NLGN1-Fc (B), LRRTM2-Fc (D), or Slitrk2-Fc (F) in the indicated transfection and treatment conditions. Neurons were analyzed at 21–24 DIV. n = 30 cells for each condition from three independent experiments, one-way ANOVA, $P < 0.0001$. ***$P < 0.001$, **$P < 0.01$, N.S., and not significant by Tukey's multiple comparisons test. Data are presented as mean ± SEM.

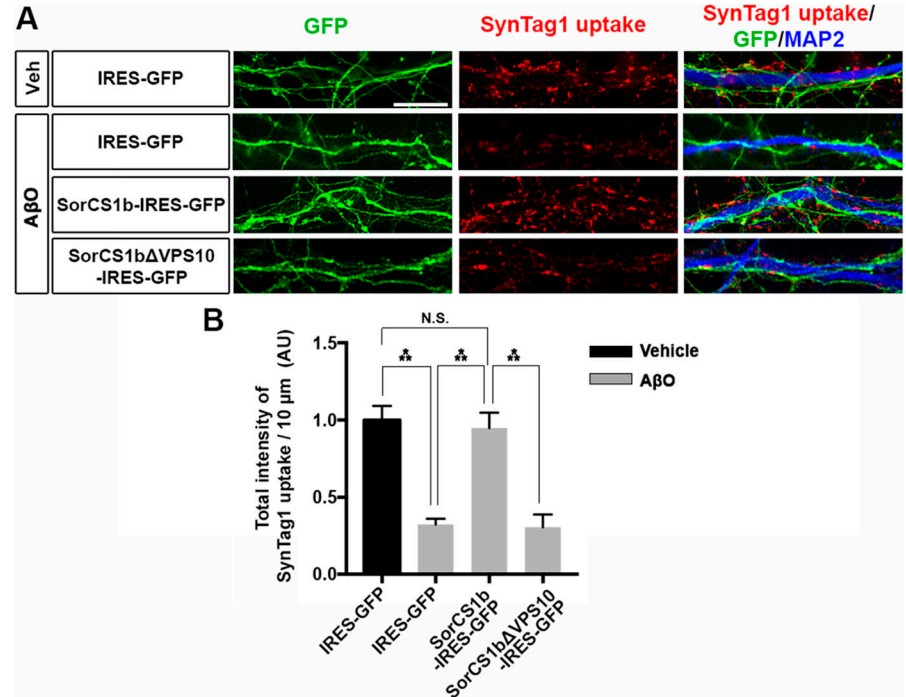

**Figure 5. Axonal expression of SorCS1b rescues AβO-induced impairment of presynaptic vesicle recycling.**
**(A)** Representative images of the uptake of anti–synaptotagmin-1 luminal antibody (SynTag1) in live cultured hippocampal neurons transfected with IRES-GFP, SorCS1b-IRES-GFP, or SorCS1bΔVPS10-IRES-GFP after a 24-h treatment with AβOs (500 nM, monomer equivalent) or vehicle (Veh). At the end of the 30-min uptake incubation, the neurons were fixed and immunostained to detect the internalized SynTag1 antibody and MAP2. Scale bar: 10 µm. **(B)** Quantification of the total intensity of SynTag1 uptake per 10 $\mu$m dendrite length in the presence and absence of AβOs. Neurons were analyzed at 21–24 DIV. n = 30 cells for each condition from three independent experiments, one-way ANOVA, $P < 0.0001$, ***$P < 0.001$, N.S., not significant by Tukey's multiple comparisons test. Data are presented as mean ± SEM.

dendritic, SorCS1. The transfected neurons were treated for 24 h with 500 nM AβOs or vehicle as a negative control before SynTag1 uptake assays. We first confirmed that AβO treatment significantly diminished SynTag1 antibody uptake in axons expressing only GFP, consistent with previous studies (Kelly & Ferreira, 2007; Park et al, 2013). Interestingly, SynTag1 antibody uptake in AβO-treated axons expressing SorCS1b was comparable to that in vehicle-treated axons expressing GFP alone, suggesting that exogenous SorCS1 expression in axons normalizes AβO-induced impaired synaptic vesicle recycling (Fig 5). On the other hand, the level of SynTag1 antibody uptake in AβO-treated axons expressing SorCS1bΔVPS10 was comparable to that in AβO-treated axons expressing only GFP, suggesting that preventing SorCS1 interaction with β-NRXs eliminates the ability of SorCS1 to normalize AβO-induced impaired synaptic vesicle recycling (Fig 5). These findings suggest that axonal SorCS1b expression rescues AβO-induced impaired synaptic vesicle recycling through cis-interaction with axonal β-NRXs.

### SorCS1 expression in axons rescues AβO-induced structural defects in excitatory synapses

In addition to impairing presynaptic differentiation and function (Figs 4 and 5), AβO treatment induces loss of excitatory synapses associated with thinning of postsynaptic density (PSD) and down-regulation of PSD-95 (Roselli et al, 2005; Shankar et al, 2007; Roselli et al, 2009; Wei et al, 2010). Given that presynaptic NRXs regulate both pre- and postsynaptic organization through trans-interactions with multiple postsynaptic organizers including NLGNs and LRRTMs, which bind to PSD-95 (Irie et al, 1997; Sudhof, 2008, 2017; Linhoff et al, 2009; Gomez et al, 2021), we next investigated whether exogenous SorCS1b expression in axons could also rescue AβO-induced excitatory synapse loss and structural changes of pre- and post-synaptic sites. To do so, we assessed the synapse density by immunostaining for the excitatory pre- and postsynaptic markers VGLUT1 and PSD-95, respectively, after AβO treatment in hippocampal neurons expressing SorCS1b and GFP or GFP alone (Fig 6). The density of excitatory synapses was measured as the number of VGLUT1-positive PSD-95 puncta per dendrite length. To investigate the effect of axonal, but not dendritic, SorCS1b in excitatory synapses, we imaged and analyzed non-transfected (GFP-negative) dendrites innervated by multiple transfected (GFP-positive) axons. In these dendrites innervated by axons expressing GFP alone, AβO treatment significantly reduced the density of excitatory synapses compared to the vehicle-treated condition (Fig 6A and B). In addition, AβO treatment significantly reduced the size of both VGLUT1 (Fig 6C) and PSD-95 (Fig 6D) puncta compared to the vehicle-treated condition. Thus, AβOs induced loss of excitatory synapses accompanied by a significant shrinkage of excitatory presynaptic sites and postsynaptic densities. In contrast, AβO treatment failed to reduce the excitatory synapse density and the size of VGLUT1 and PSD-95 puncta in non-transfected dendrites innervated by multiple SorCS1b-transfected axons, with these measures being comparable to those in vehicle-treated dendrites innervated with axons expressing only GFP (Fig 6). These findings suggest that axonal SorCS1b expression normalizes the AβO-induced structural defects in excitatory synapses including the loss of excitatory synapses and the shrinkage of excitatory presynaptic sites and postsynaptic densities.

## Discussion

In this study, we explored how binding between SorCS1 and β-NRXs impacts synapses upon AβO exposure. We defined the HRD of β-NRXs as the domain responsible for the interaction between the SorCS1/2 ectodomains and β-NRXs and found that the SorCS1 ectodomain and AβOs compete for binding to NRX1β. Furthermore, SorCS1b colocalizes with NRX1b on the axon surface, especially at axon-dendrite contact sites through the ectodomain interaction. Notably, axonal SorCS1b expression normalizes several AβO-induced synaptic pathologies, preventing the impairment of NRX-mediated presynaptic organization and restoring synaptic vesicle recycling and excitatory synapse structure. Thus, we propose that SorCS1b plays a beneficial role in alleviating AβO-induced synaptic pathology by competing with AβOs for β-NRX binding on axons.

One of the important findings of this study is that the SorCS1 ectodomain binds to NRX1β via its HRD and competes against AβOs for binding to NRX1β. The HRD is an N-terminal domain unique to β-NRXs, and very little was previously known about its function. Given that the SorCS1 ectodomain is common to all SorCS1 isoforms (Hermey, 2009), our finding suggest that the HRD could be a key determinant by which SorCS1 recognizes NRX1β as a protein target of the sorting receptor. A study has proposed that SorCS1 expressed on the cell membrane interacts with NRX1β in a cis, but not a trans, manner (Savas et al, 2015). Consistent with this, our cell surface–binding experiments show that AβO-NRX1β binding is suppressed by the co-expression of SorCS1b and NRX1β in the same cell, which presumably results in cis-interaction between SorCS1b and NRX1β. Therefore, SorCS1 is likely to interfere with AβO-NRX1β binding by cis-interaction with NRX1β via the HRD. However, it remains possible that other interaction modes may be also involved in interfering with AβO-NRX1β binding because previous studies have shown that the SorCS1 ectodomain is frequently shed by metalloproteases and γ-secretases, resulting in the production of soluble SorCS1 ectodomain proteins that retain their ligand-binding ability (Hermey et al, 2006; Nyborg et al, 2006; Willnow et al, 2008). However, whether the SorCS1 ectodomain is soluble or expressed on the surface, any interaction between the SorCS1 ectodomain and NRX1β is beneficial as it shields NRX1β from AβOs. Conversely, AβOs binding to the HRD of NRX1β interfere with the SorCS1–NRX1β interaction. This would be detrimental to the normal trafficking and function of NRX1β as SorCS1 regulates the axonal transport of NRXs (Savas et al, 2015). To increase the beneficial effects of the SorCS1 ectodomain, future studies are crucial to elucidate the structural basis of the SorCS1-β–NRXHRD interaction and to determine the amino acid residues responsible for the SorCS1-β–NRXHRD and AβO-β–NRXHRD interactions. Such studies would be helpful for designing small molecules and peptides that could enhance SorCS1 ectodomain binding and/or reduce AβO binding to shield β-NRXs from AβOs for the potential rescue of AβO-induced synaptic pathology.

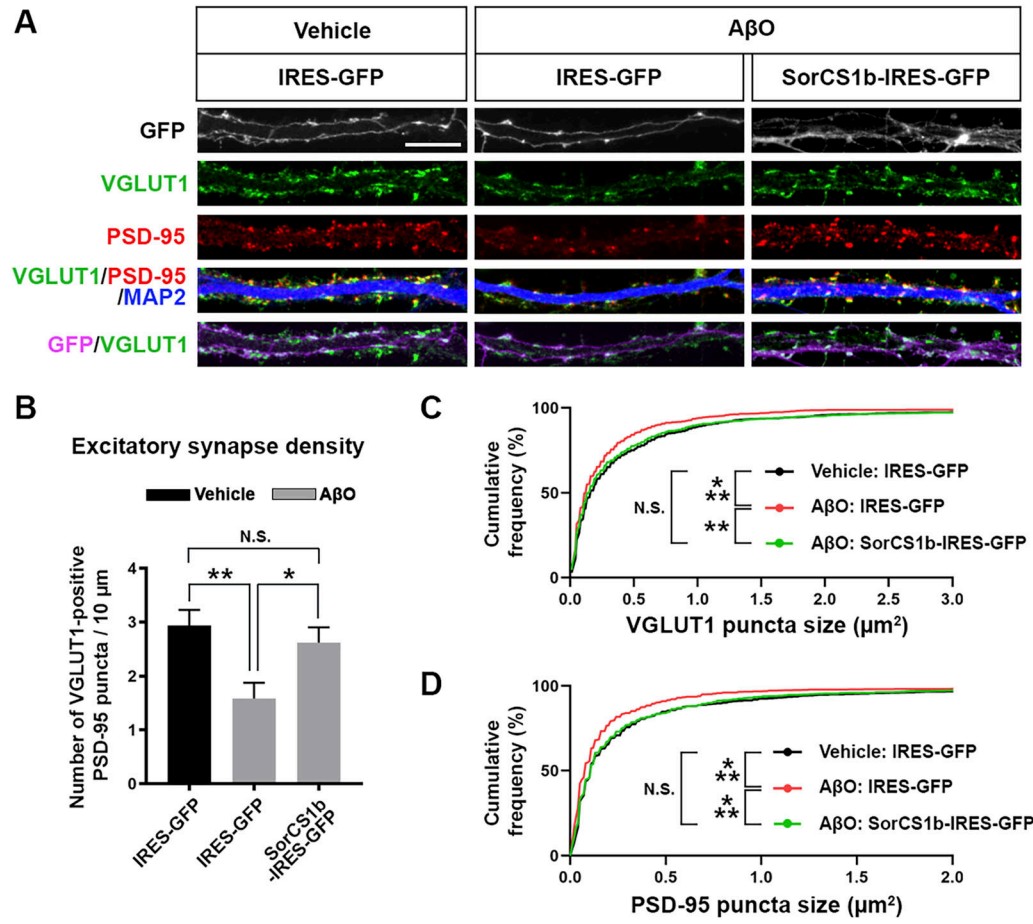

**Figure 6. Axonal expression of SorCS1b rescues AβO-induced structural changes in excitatory synapses.**
**(A)** Representative images of cultured hippocampal neurons transfected with IRES-GFP or SorCS1b-IRES-GFP and treated for 24 h with AβOs (500 nM, monomer equivalent) or vehicle at 21 DIV. After the treatment, the neurons were triple immunostained for VGLUT1, PSD-95, and MAP2 at 22 DIV. Scale bar: 10 μm. **(B)** Quantification of the density of VGLUT1-positive PSD-95 clusters as a measure of excitatory synapse density. Neurons were analyzed at 22 DIV. n = 24 cells for each condition from three independent experiments, one-way ANOVA, $P < 0.01$. **$P < 0.01$, and *$P < 0.05$, N.S., not significant by Tukey's multiple comparisons test. Data are presented as mean ± SEM. **(C, D)** Cumulative frequency distribution curves of VGLUT1 puncta size (C) and PSD-95 puncta size (D) in 24 neurons for each condition from three independent experiments. n = 1016, 917, and 1102 VGLUT1 puncta and 975, 804, and 941 PSD-95 puncta in IRES-GFP–transfected neurons with vehicle treatment (Vehicle: IRES-GFP), IRES-GFP–transfected neurons with AβO treatment (AβO: IRES-GFP), and SorCS1b-IRES-GFP–transfected neurons with AβO treatment (AβO: SorCS1b-IRES-GFP), respectively. ***$P < 0.001$, **$P < 0.01$, and N.S., not significant by Kolmogorov–Smirnov test.

Previous studies have demonstrated that SorCS1cβ is mainly localized in endosomal compartments as a sorting receptor and colocalizes with NRX1β in endosomes in HeLa cells and in dendrites when SorCS1cβ and NRX1β are co-transfected (Savas et al, 2015). Moreover, SorCS1 knockout (KO) causes mis-sorting of NRX1β to the dendritic surface and consequently decreases NRX1β on axonal surfaces, eventually resulting in its degradation (Savas et al, 2015). On the other hand, we demonstrated that SorCS1b can be expressed on the surface of COS-7 cells and on the axon surface of cultured hippocampal neurons and their dendrite surface. Moreover, our cell surface–binding assays revealed that the co-expression of full-length SorCS1b, but not of SorCS1bΔVPS10, with NRX1β suppresses AβO binding to COS-7 cells with the NRX1β surface expression. These findings suggest that SorCS1, at least the SorCS1b isoform, acts as an AβO competitor for surface β-NRX binding and, given the high colocalization between SorCS1b and NRX1β, plays a role in shielding β-NRXs from AβOs on the axon surface.

Our recent study suggested that AβOs diminish NLGN1-induced presynaptic organization by decreasing surface β-NRXs on axons (Naito et al, 2017). In the present study, our data indicate that SorCS1b is associated with the NLGN1–NRX1β trans-complex without interfering with NLGN1–NRX1β interaction. At axon–dendrite contact sites, SorCS1b co-clusters with NRX1β on the axon surface. Furthermore, the axonal SorCS1b expression rescues not only AβO-induced impairment of NLGN1-induced excitatory presynaptic differentiation but also AβO-induced PSD shrinkage. Together, we propose that under Aβ pathological conditions, SorCS1b functions as a unique stabilizer for the trans-synaptic complex of axonal β-NRXs and dendritic NLGNs through cis-interaction with β-NRXs and that this stabilizing effect of SorCS1b contributes to the prevention of AβO-induced impaired organization of both presynaptic and postsynaptic sites. In addition, SorCS1b had no effect on RPTP-based presynaptic organization activity and no binding ability to any RPTPs, suggesting a specific beneficial role of SorCS1 in stabilizing NRX-based synaptic organizing complexes.

Another important finding of the present study is that the axonal expression of SorCS1b, but not SorCS1bΔVPS10, which has no binding to NRX1β, rescues AβO-induced impairment of synaptic vesicle recycling. Although α-NRXs and β-NRXs both regulate synaptic release through distinct mechanisms impinging on pre-synaptic calcium regulation (Missler et al, 2003; Sudhof, 2008, 2017; Anderson et al, 2015), it remains unclear whether and how AβO–NRX interaction is involved in AβO-induced impaired synaptic release. Our previous study has suggested that AβO-β–NRX interaction down-regulates functional β-NRXs at presynaptic terminals by reducing surface β-NRXs on axons (Naito et al, 2017). Importantly, conditional β-NRX triple-KO in cultured hippocampal neurons exhibited impaired excitatory synapse release through endo-cannabinoid (EC) signaling (Anderson et al, 2015). AβO treatment in vitro and in vivo and overproducing Aβ in AD mouse lines also result in changes in molecules linked to EC signaling (Mulder et al, 2011; Orr et al, 2014). These findings suggest that the AβO-induced re-duction of surface β-NRXs followed by changes in EC signaling could be a potential key mechanism underlying AβO-induced impairment of excitatory synaptic vesicle recycling and gluta-mate release. Whether and how SorCS1 influences this EC-signaling mechanism under both normal physiological and AD pathological conditions will be important to explore in future studies.

Another pertinent question is how SorCS1b rescues AβO-induced presynaptic pathologies such as synaptic vesicle recycling im-pairment and VGLUT1 puncta shrinkage. We propose that SorCS1 interacting with β-NRX HRD may shield β-NRXs from AβO-induced dysfunction. However, SorCS1 can also regulate axonal surface polarization of α-NRXs via Rab11 (Ribeiro et al, 2019), which must be based on HRD-independent mechanisms because α-NRXs do not possess an HRD (Reissner et al, 2013). Although the SorCS1 ecto-domain did not bind to any isoforms of α-NRXs, it is possible that SorCS1 rescues AβO-induced impaired synaptic recycling through both α-NRXs and β-NRXs but through two distinct molecular mechanisms: one being an HRD-independent mechanism involving axonal polarization of surface α-NRXs via Rab11, and the other, an HRD-dependent mechanism involving SorCS1-β–NRX binding that inhibits AβO-β–NRX binding.

To date, a few studies have addressed the presynaptic mech-anisms of AβO pathology, implicating NRXs (Brito-Moreira et al, 2017; Naito et al, 2017) and a Na+/K+-ATPaseα3 subunit (Ohnishi et al, 2015). In the latter study, AD patient-derived AβOs induced presynaptic calcium overload and neuronal cell death through their binding to Na+/K+-ATPaseα3, suggesting that alleviating presynaptic calcium dysregulation in AβO pathology would be therapeutically impor-tant. NRX down-regulation has also been linked to neuronal cell death: the deletion of all NRXs from cerebellar granule cells causes significant cerebellar granule cells death, in part, depending on BDNF-TrkB neurotrophin signaling (Uemura et al, 2022). Intriguingly, SorCS1 binds to TrkB and regulates the BDNF-TrkB signaling pathway (Subkhangulova et al, 2018), and our previous study using 6-mo-old AD model mice with Aβ overproduction (J20 mouse line) showed down-regulation of NRXs in cortical and hippocampal synaptosomes (Naito et al, 2017). Thus, in a future study it will be important to address whether and how NRXs, SorCS1, and TrkB are linked to influence not only synaptic pathology but also neuronal cell death in Aβ-induced pathology in AD.

In this study, based on in vitro experiments, we assessed rela-tively acute effects of AβOs (24 h). In the future, it will be necessary to determine whether and how SorCS1 affects synaptic pathology induced by long-term exposure of neurons to AβOs, on a time scale that relates to AD progression in vivo and in patients. Additional future studies should address whether SorCS1, under in vivo conditions, can rescue synaptic pathology and neuronal damage, such as AβO-induced neuronal cell loss. To address this question, it will be necessary to create an inducible SorCS1-overexpressing mouse line and then cross it with AD model mice or to establish a viral delivery system for the SorCS1 gene for its injection into AD mouse brains. Using these mice will allow us to test whether and how neuronal SorCS1 overexpression can rescue synaptic and/or neuronal pathologies and cognitive dysfunction in vivo both before and even after the onset of Aβ pathology. As a protein-sorting receptor, SorCS1 targets not only NRXs (Savas et al, 2015; Ribeiro et al, 2019) but also other molecules related to AD pathology and synaptic and neuronal functions such as APP itself (Lane et al, 2010; Reitz et al, 2011; Hermey et al, 2015), AMPA-type glutamate receptor (Savas et al, 2015) and the neurotrophin receptor TrkB (Subkhangulova et al, 2018). Therefore, if in vivo SorCS1 over-expression rescues AD pathology and cognitive deficits, it will be important to dissect which SorCS1 targets are involved in each in vivo rescue effect, and identifying these could provide multiple avenues to investigate the development of novel therapeutic strategies for AD.

## Materials and Methods

### Plasmids

The constructs for expressing SorCS1 or SorCS2 ectodomain fused to human IgG Fc were generated by subcloning the coding se-quence of the mature forms of mouse SorCS1 ectodomain (amino acids [aa] 34–1098) and mouse SorCS2 ectodomain (aa 52–1077), respectively, into the pc4-spNRX1β-Fc cloning vector (Takahashi et al, 2011; Takahashi et al, 2012; Naito et al, 2017) after the NRX1β signal sequence (spNRX1β). pcDNA4-mouse SorCS1b-myc, which ex-presses intracellularly myc-tagged SorCS1b (SorCS1b-myc; kindly provided by Dr. Nabil Seidah), and pCpGfree-vitroNmcs-mSorCS2-WT (kindly provided by Dr. Camilla Gustafsen) were used as PCR templates for the subcloning. To generate the construct encoding SorCS1b-myc lacking its VPS10 domain (pc4-SorCS1bΔVPS10-myc), inverse PCR was performed to remove the VPS10 region (aa 196–796) in frame from pcDNA4-SorCS1b-myc. Then, to make the constructs expressing SorCS1b-myc and SorCS1bΔVPS10-myc under the control of the CAG promoter (pCAG-SorCS1b-myc and pCAG-SorCS1bΔVPS10-myc, respectively), the coding se-quences of SorCS1b-myc and SorCS1bΔVPS10-myc including the stop codons were subcloned between two EcoRI sites into pCAG-HA-NRX1βS4(+) (Kindly provided by Dr. Takeshi Uemura [Uemura et al, 2010]) by replacing the N-terminal regions of HA-NRX1βS4(+) open-reading frame. For the internal ribosome entry site (IRES)-based bicistronic constructs co-expressing GFP with either untagged full-length SorCS1b or untagged SorCS1bΔVPS10

under the same CAG promoter (pCAG-SorCS1b-IRES-GFP and pCAG-SorCS1bΔVPS10-IRES-GFP, respectively), the cloning vector pCAG-IRES-GFP was first generated by subcloning the sequence of IRES followed by GFP (IRES-GFP) into pCAG-GFP (kindly provided by Dr. Connie Cepko through Addgene [Matsuda & Cepko, 2007]) between EcoRI and NotI, thus replacing GFP with IRES-GFP. Next, the coding sequences of SorCS1b and SorCS1bΔVPS10 including the stop codons, but excluding the myc coding sequences, were amplified by PCR using their respective pcDNA4-myc constructs as templates, and the products were sub-cloned into pCAG-IRES-GFP at the EcoRI site. To make the construct expressing the NRX1β ectodomain lacking the HRD fused to Fc (pc4-NRX1βΔHRD-Fc), the coding sequence of the mature form of the NRX1βS4(−) ectodomain lacking its HRD (aa 55–83) was amplified by PCR using HA-NRX1βΔHRD as a template and then sub-cloned into pc4-spNRX1β-Fc cloning vector after the NRX1β signal sequence. The following plasmids were kind gifts: pCAG-HA-NRX1βS4(−) and pCAG-HA-NRX1βS4(+) from Dr. Takeshi Uemura (Shinshu University); HA-NLGN1A(−)B(−) and NRX1βS4(−)-Fc from Dr. Peter Scheiffele (University of Basel) via Addgene (Scheiffele et al, 2000); HA-NLGN2, YFP-NLGN3, and LRRTM2-CFP from Dr. Ann Marie Craig (University of British Columbia); LAR-CFP from Dr. Eunjoon Kim (Korea Advanced Institute of Science and Technology); HA-glutamate receptor delta-1 (GluD1) and HA-GluD2 from Dr. Michisuke Yuzaki (Keio University); and IL1RAPL1-pFLAG and IL1RAcP-pFLAG from Dr. Tomoyuki Yoshida (Toyama University). The other constructs used in this study, including a series of extracellularly tagged HA-NRXs and NRXs lacking their HRDs, NLGN1-Fc, LRRTM2-Fc, and so on, were described previously (Naito et al, 2017). All constructs were verified by DNA sequencing.

## Aβ preparation

Aβ (1–42) (Cat# A-1002-2, 1 mg, r-peptide) and biotin-tagged Aβ (1–42) (Cat# AS-23523-05, 0.5 mg, Anaspec) were used to make oligomeric forms, as we did previously (Naito et al, 2017) based on the method described in an earlier study (Stine et al, 2003). The preparations were aliquoted and stored at −80°C or used in experiments immediately. Individual AβO stocks were never thawed and refrozen. Briefly, lyophilized peptides were dissolved in 1,1,1,3,3,3-hexafluoro-2-propanol (HFIP, Cat# 52517; Millipore Sigma) to ensure that the starting material was in a homogenous, non-aggregated monomeric state, and then, aliquots of this solution were placed in tubes for 2 h at RT. The HFIP was then evaporated in a vacuum centrifuge concentrator (SPD131; Thermo Fisher Scientific) yielding Aβ peptide films. Before use, each peptide film was reconstituted in dimethyl sulfoxide (DMSO, Hybri-Max Cat# D2650; Millipore Sigma) to obtain a 1 mM Aβ stock solution, which was then incubated in a bath sonicator for 10 min. The peptide stock was then diluted to 100 μM with 10 mM Tris–HCl (pH 7.4) and incubated for 48 h at 22°C to facilitate the formation of oligomers of higher molecular weight.

## Production of soluble Fc-fusion proteins and cell surface–binding assays

SorCS1-Fc, SorCS2-Fc, NRX1β-Fc, NRX1βΔHRD-Fc, NLGN1-Fc, and Fc (a negative control) were generated using HEK293T cells transfected

with the corresponding expression vectors using TransIT-PRO Transfection Reagent (Cat# MIR5740; Mirus Bio) and maintained in a serum-free AIM V synthetic medium (Cat# 12055083; Thermo Fisher Scientific) for 3 d, and then purified from this culture medium using Protein G Sepharose beads (Cat# GE17-0618-01; Millipore Sigma), as described previously (Takahashi et al, 2011; Takahashi et al, 2012; Naito et al, 2017). To test for interaction of the Fc-fused recombinant proteins or biotin-AβOs with our proteins of interest, including NRXs, COS-7 cells cultured on coverslips were transfected with the indicated expression vectors using TransIT-PRO Transfection Reagent and maintained for 24 h. The transfected COS-7 cells were washed with an extracellular solution (ECS) containing 2.4 mM KCl, 2mM CaCl$_2$, 1.3mM MgCl$_2$, 168 mM NaCl, 20 mM HEPES (pH 7.4), and 10 mM D-glucose with 100 μg/ml of BSA (ECS/BSA). Next, the transfected COS-7 cells were incubated with Fc-fused recombinant proteins and/or biotin-AβOs in ECS/BSA for 1 h at 4°C to prevent endocytosis. The cells were washed three times using ECS, then fixed using a parafix solution (4% paraformaldehyde and 4% sucrose in PBS [pH 7.4]) for 12 min at RT. To label surface HA, bound Fc proteins and/or bound biotin-AβOs, the fixed cells were then incubated with a blocking solution (PBS + 3% BSA and 5% normal donkey serum) for 1 h at RT. Afterward, without cell permeabilization, they were incubated with primary antibodies in the blocking solution overnight at 4°C with secondary antibodies and/or fluorescent-conjugated streptavidin for 1 h at RT. To label total myc together, the fixed cells were permeabilized with PBST (PBS + 0.2% Triton X-100) after labelling surface HA. The following primary antibodies were used for immunocytochemistry: anti-HA (1:2,000; rabbit IgG, Cat# ab9110; Abcam) and anti-myc (1:2,000; mouse IgG1, Cat# sc-40; Santa Cruz). The following highly cross-absorbed, Alexa dye-conjugated or AMCA-conjugated secondary antibodies (1:500; Jackson ImmunoResearch) were used: donkey Alexa488-conjugated anti-rabbit IgG (H+L), donkey Alexa647-conjugated anti-mouse IgG (H+L), donkey Alexa594-conjugated anti-human IgG (H+L), and donkey AMCA-conjugated anti-human IgG (H+L). To label bound biotin-AβOs, Alexa594-conjugated streptavidin or AMCA-conjugated streptavidin (1:4,000; Jackson ImmunoResearch) was used.

## Pull-down assays

Recombinant NRX1β-Fc and NRX1βΔHRD-Fc proteins were pre-immobilized with Protein G Magnetic Beads (Dynabeads Protein G, Cat# 10004D; Thermo Fisher Scientific) in PBS overnight at 4°C with constant agitation. The pre-immobilized beads were then incubated with 50 nM recombinant mouse SorCS1 ectodomain tagged with a C-terminal 6-His tag (SorCS1-His, Cat# 4395-SR-050; R&D systems) in ECS for 2 h at 4°C. The bead–protein complexes were isolated by using a magnetic stand (DynaMag-2 magnet, Cat# 12321D; Thermo Fisher Scientific) to isolate the pull-down fraction, and the supernatant was collected as the unbound fraction. The isolated bead complexes were then washed three times with an ECS solution. The proteins bound on beads were eluted by boiling in SDS sample buffer containing β-mercaptoethanol, separated by SDS–PAGE, and analyzed using Western blotting. Anti-SorCS1 (1:1,000, Rabbit, Cat# ab93331; Abcam) and anti-His tag (1:2,000; mouse IgG2a, clone OGHis, Cat# D291-3; MBL) primary antibodies and

donkey HRP-conjugated anti-rabbit IgG (H+L) and donkey HRP-conjugated anti-mouse IgG (H+L) (1:2,000; Jackson Immuno-Research) secondary antibodies were used to detect the bound SorCS1-His in the pull-down fraction and the applied SorCS1-His in the unbound fraction, respectively. To detect the immobilized Fc proteins, HRP-conjugated anti-human IgG (H+L) antibody (1:2,000; Jackson ImmunoResearch) was used. In this assay, we observed an unexpected slower migration in SDS–PAGE of NRX1βΔHRD-Fc than NRX1β-Fc (Fig 1D), which is presumably a "gel-shifting" phenomenon (Rath et al, 2009). Indeed, we further validated the NRX1βΔHRD-Fc protein as lacking the HRD but retaining NLGN1 binding ability by Western blotting using an anti-NRX1β antibody that recognizes an epitope in the HRD (clone N170A/1; Neuromab) and cell surface–binding assays using COS-7 cells expressing HA-NLGN1/2, respectively (Fig S2).

### Neuron culture, transfection, and neuronal immunocytochemistry

Primary rat hippocampal neuron cultures were prepared from embryonic day 18 (E18) rat embryos as described previously (Kaech & Banker, 2006). All animal experiments were carried out in accordance with the Canadian Council on Animal Care guidelines and approved by the *Institut de recherches cliniques de Montréal* (IRCM) Animal Care Committee. Transfection into hippocampal neurons was performed using the AMAXA nucleofector system (Lit, VPG-1003; Program: O-003; Lonza) before plating the dissociated hippocampal cells onto coverslips (0 d in vitro [DIV]). At the end of the experiment, neurons were fixed with parafix solution for 12 min, permeabilized with PBST (except for experiments examining surface expression), and then blocked with the blocking solution. Afterward, they were incubated with primary antibodies in the blocking solution overnight at 4°C and with secondary antibodies for 1 h at RT. To label surface SorCS1 and/or HA-NRX1β together with MAP2, the fixed neurons were incubated with primary antibodies against SorCS1 and/or HA without cell permeabilization and then permeabilized with PBST for MAP2 immunostaining. The following primary antibodies were used for immunocytochemistry: anti-SorCS1 antibody (1:1,000, Rabbit, Cat# ab93331; Abcam), anti-VGLUT1 (1:1,000; guinea pig; Cat# AB5905; Millipore Sigma), anti-PSD-95 (1:500; mouse IgG2a; clone 6G6-1C9, Cat# MA1-045; Thermo Fisher Scientific), anti-MAP2 (1:2,000; chicken polyclonal IgY; Cat# ab5392; Abcam), and anti-HA (1:1,000; mouse IgG1; clone HA-7, Cat# H3663). Highly cross-adsorbed, Alexa dye-conjugated secondary antibodies generated in donkeys toward the appropriate species (1:500; Alexa488, Alexa594, and Alexa647; Jackson ImmunoResearch) were used as detection antibodies.

### Protein clustering assays and artificial synapse formation assays

Protein G-coated Magnetic Beads (Dynabeads Protein G, Cat# 10004D; Thermo Fisher Scientific) were incubated with recombinant NLGN1-Fc, LRRTM2-Fc, Slitrk2-Fc or Fc (a negative control) in PBS + 3% BSA (PBSA) overnight at 4°C. After they were washed with PBSA using a magnetic stand (DynaMag-2 magnet, Cat# 12321D; Thermo Fisher Scientific), the coated beads were resuspended in conditioned neuronal culture media and applied to 20–23 DIV hippocampal neurons transfected with the indicated constructs. For

protein-clustering assays, the neurons were maintained for 6 h and fixed with a parafix solution for immunocytochemistry. For artificial synapse formation assays, AβOs (500 nM monomer equivalent) or Tris–HCl (50 µM, pH 7.4; vehicle control) were added simultaneously with the beads to the culture media. The neurons were then maintained for 24 h and fixed with a parafix solution for immunocytochemistry.

### SynTag1 antibody uptake assays

For assessing changes in vesicle recycling rate at presynaptic terminals, SynTag1 antibody uptake assays were conducted as described previously (Ammendrup-Johnsen et al, 2015). First, neurons were incubated with AβOs (500 nM, monomer equivalent) or vehicle control at 20–23 DIV for 24 h before the SynTag1 uptake experiments. On the following day, the AβO- or vehicle-treated live neurons were incubated with an antibody recognizing the luminal domain of SynTag1 (1:500; mouse IgG1, clone 604.2, Cat# 105 311; Synaptic Systems) for 30 min in a culture medium at 37°C in a 5% $CO_2$ incubator. The neurons were washed with culture media three times and fixed with a parafix solution for immunocytochemistry.

### Imaging and quantitative fluorescence analysis

For quantitative analysis, all image acquisitions, analyses, and quantifications were conducted by investigators blinded to the experimental conditions. Cell culture images were acquired on a Leica DM6000 fluorescent microscope with a 40 × 0.75 NA dry objective or 63 × 1.4 NA oil objective and a Hamamatsu cooled CCD camera using Volocity software (Perkin Elmer). Images were obtained in 12-bit grayscale and prepared for presentation using Adobe Photoshop 2020. For quantification, sets of cells were immunostained simultaneously and imaged with identical microscope settings. Analysis for the cell surface–binding assay was performed using Volocity, and that for the other assays was performed using Metamorph 7.8 (Molecular Devices). For the cell surface–binding assays, after off-cell background intensity was subtracted, the average intensity of bound proteins per COS-7 cell region was measured and normalized to the average intensity of the surface HA signal. The half-maximal inhibitory concentration ($IC_{50}$ value) was determined by non-linear regression curve fit in GraphPad Prism 9 (GraphPad Software). For colocalization assays, axon co-expressing surface SorCS1b constructs and surface HA-NRX1β constructs were traced by a multi-line command, and their average intensity values along the selected line regions were measured by a linescan command. Then, the Pearson correlation coefficient values were calculated as colocalization values in GraphPad Prism 9. For the artificial synapse formation assays, Fc protein-coated beads contacting transfected axons were selected based on phase contrast images and GFP and MAP2 fluorescent images. The VGLUT1 images were thresholded, and the average intensity of VGLUT1 puncta within concentric circular regions measuring 1.5-times the diameter of the beads was measured. For protein-clustering assays, Fc protein-coated beads contacting axons co-expressing SorCS1 and HA-NRX1β were selected based on phase contrast images and all fluorescent images. The average intensity of surface SorCS1 and HA signals within concentric circular

regions measuring 1.5-times the diameter of the beads was measured. For the SynTag1 uptake assays and excitatory synapse number analysis, non-transfected dendrites innervated by multiple GFP-positive transfected axons were first selected to investigate the effects of SorCS1 expression on axons. Then, for the SynTag1 uptake assays, the SynTag1 channel was thresholded to extract SynTag1 puncta, and their total intensity per dendrite length was measured. For excitatory synapse number analysis, VGLUT1 and PSD-95 channels were thresholded to isolate the puncta, and the number of PSD-95 puncta overlapping with VGLUT1 puncta per dendrite length was measured. For the VGLUT1 and PSD-95 cluster size measurement, VGLUT1 and PSD-95 channels were thresholded to isolate the puncta, and the size of each cluster was analyzed.

### Statistical analysis

Statistical tests were performed using GraphPad Prism 9 (GraphPad Software). The data distribution was assumed to be normal, but this was not formally tested. Statistical comparisons were performed using one-way ANOVA with post hoc Tukey's multiple comparisons tests, ANOVA with post hoc Dunnett's tests, Kolmogorov–Smirnov tests, or $t$ tests as indicated in each figure legend. Data were obtained from three independent experiments and statistical significance was defined as $P < 0.05$.

## Data Availability

This study includes no data deposited in external repositories.

## Supplementary Information

## Acknowledgments

We thank Nicolas Chofflet for his technical assistance. This study was supported by a Canadian Institutes of Health Research (CIHR) Project Grant (PTJ-159588), an Alzheimer Society Research Program (ASRP) Biomedical Research Grant (18-03), a Natural Science and Engineering Research Council (NSERC) Discovery Grant (RGPIN-2017-04753), and Fonds de la recherche du Québec Research Scholars (Senior-251655) to H Takahashi, and an NSERC doctoral award, an FRQS doctoral award (252652), an IRCM doctoral scholarship and a McGill medical internal studentship to AK Lee, ASRP Doctoral scholarship (23-12) to N Yi and an FRQS doctoral award (303256) to H Khaled.

### Author Contributions

AK Lee: formal analysis, validation, visualization, methodology, and writing—original draft, review, and editing.
N Yi: investigation and writing—review and editing.
H Khaled: resources, validation, investigation, visualization, methodology, and writing—review and editing.
B Feller: formal analysis, validation, investigation, and writing—review and editing.

H Takahashi: conceptualization, resources, formal analysis, supervision, funding acquisition, validation, investigation, visualization, methodology, project administration, and writing—original draft, review, and editing.

### Conflict of Interest Statement

The authors declare that they have no conflict of interest.

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
