## [Reviewer comments · Life Science Alliance]

Life Science Alliance

SorCS1 inhibits amyloid- β binding to neurexin and rescues amyloid- β -induced synaptic pathology

Alfred Lee, Nayoung Yi, Husam Khaled, Benjamin Feller, and Hideto Takahashi

DOI: <https://doi.org/10.26508/lsa.202201681>

Corresponding author(s): *Hideto Takahashi, Montreal Clinical Research Institute*

Review Timeline:

Submission Date:	2022-08-18
Editorial Decision:	2022-09-26
Revision Received:	2022-12-23
Editorial Decision:	2023-01-11
Revision Received:	2023-01-13
Accepted:	2023-01-16

Scientific Editor: *Eric Sawey, PhD*

Transaction Report:

September 26, 2022

Re: Life Science Alliance manuscript #LSA-2022-01681-T

Dr. Hideto Takahashi
Institut de Recherches Cliniques de Montréal, Montreal
Canada

Dear Dr. Takahashi,

Thank you for submitting your manuscript entitled "SorCS1 inhibits amyloid- β binding to neuexin and rescues amyloid- β -induced synaptic pathology" to Life Science Alliance. The manuscript was assessed by expert reviewers, whose comments are appended to this letter. We invite you to submit a revised manuscript addressing the Reviewer comments.

Thank you for this interesting contribution to Life Science Alliance. We are looking forward to receiving your revised manuscript.

Sincerely,

B. MANUSCRIPT ORGANIZATION AND FORMATTING:

Reviewer #1 (Comments to the Authors (Required)):

REVIEW - AK LEE ET AL

GENERAL COMMENTS

This is a top 10% piece of research. It concerns a topic that is significant and should be of wide interest, namely, the mechanism of action by which the synaptotoxicity of amyloid beta oligomers (A β O) is mediated. The issue is relevant to the cause of Alzheimer's disease and presents the possibility of a novel therapeutic target. Results reveal important molecular details regarding the toxic interaction of A β O with neurexin and shows the competitive cis action of SorCS1 in blocking this interaction. The data are of high quality and support the conclusion. The findings are supported by excellent writing, which clearly states the problem and its solution. The flow of the paper is logical, and each paragraph communicates its message effectively. The methods are described in detail. Overall, this manuscript is a model of matching conclusions with evidence, of good organization, and of clarity.

ESSENTIAL CHANGE - Fix the Scholarship

- Simple but significant emendation is needed before I can recommend this manuscript be accepted - the time and effort required are minimal. The manuscript concerns the mechanism of synapse damage caused by amyloid beta oligomers (A β O). A β O are neurotoxins widely considered to play a pathogenic role in Alzheimer's disease. The study is introduced in a scholarly manner. The authors credit numerous earlier works for establishing the foundation to the current investigation. Their scholarship is not adequate, however, and seminal work has been neglected, despite it being well known and highly germane to this study. These salient publications also were ignored in an excellent earlier study by this group, published in Scientific Reports. This gap in scholarship should be corrected before the manuscript is published.
- The lead paragraph covers the cogent milestones that define the field. The milestones are well-chosen and set the stage for the current work. They appear in the following order.
(1) Accumulation of "toxic A β peptides" is a characteristic of AD ... major milestones are then reviewed, and the authors cite studies linking A β O to the following neural damage: (2) Synapse dysfunction and loss; (3) Decreases in pre-/postsynaptic proteins; (4) Decreases in synaptic vesicle recycling; (5) Decreases in synaptic spines; (6) Impaired LTP; (7) Enhanced LTD; (8) In vivo substantiation of A β O impact on synaptic protein expression, dendritic spine density, and LTP.
- Based on the cited evidence, the paragraph concludes "...synapses are vulnerable to A β , and understanding the molecular mechanism that underlie this vulnerability is crucial for explain how A β induces synapse pathology and how A β -induced synapse pathology could be ameliorated."
- In reviewing the literature, the authors have missed a set of benchmark studies that need to be added. These seminal studies are well known and germane to the milestones of the first paragraph. Point by point:
(1) Accumulation of toxic peptides in AD ... The "toxic" A β peptides are A β O, and the seminal works regarding accumulation of A β O in AD are Kaye et al, Science, 2003, and Gong et al, PNAS, 2003. (2,3,5) Synapse dysfunction and loss and decreases in pre-/postsynaptic proteins. Missing seminal studies, which showed that A β O target synapses, cause loss of key synaptic proteins, and induce spine degeneration are Lacor et al, J. Neurosci., 2004, 2007. (6) Impaired LTP. The missing study, which also introduced the hypothesis that A β O cause the neural damage leading to AD, is Lambert et al, PNAS, 1998. (7) Enhanced LTD. The missing study is Wang et al, Brain Res, 2002.
- Given their scholarly overview, the authors should cite papers in which the A β O hypothesis was introduced (Lambert et al, PNAS, 1998) and cogently reviewed (Cline et al, JAD, 2018). They also should include the influential (but somewhat controversial) paper on A β O-induced memory loss (Lesné et al, Nature, 2006).

MINOR POINTS

- A graphic illustrating their mechanism would be appealing.
- It could be useful for the authors to comment on the work by Ohnishi et al, PNAS 2015, which attributed the presynaptic mechanism of A β O to their binding to the Alpha3-NaK ATPase.
- The method for preparing A β O cites their earlier paper (Naito et al) but it appears to be based on the method introduced by Stine et al, JBC, 2003, which they should cite.
- The authors (and the field) should avoid referring to "A β " and to "amyloid" when discussing the action of "A β O." The lead

paragraph, e.g., states "A β induces synaptic pathology." As the authors make clear, it is not "A β " that induces the pathology - it is A β oligomers (A β Os). The field's literature suffers from incorrect, interchangeable, indiscriminate use of amyloid, A β , and A β Os.

- It might be useful to comment on limits of therapeutic benefits derivable from targeting the SorCS1/neurexin mechanism. For example, would targeting this mechanism help protect against other effects listed in their introduction (e.g., pathological removal of post-synaptic proteins or dysfunction in synaptic plasticity)?
- A section on abbreviations would be helpful.
- Occasionally, sentences are too long and complex.

Reviewer #2 (Comments to the Authors (Required)):

This very interesting and well-conducted study nicely demonstrates a protective role for SorCS1 with respect to amyloid-beta oligomers induced synaptic pathology. The paper is well written and the conclusions made by the authors are for the most part supported by the data, without over-interpretations.

However, some issues need to be addressed before the paper can be considered for publication:

- The authors should provide some quality control data (as they did in Naito et al., 2017) with respect to the amyloid-beta oligomers preparations they used in this study. Some of the experiments, such as the amyloid-beta/SorCS1 competition for NRX1beta binding (Fig 2) would be strengthened by including a non-oligomeric amyloid-beta "fresh" sample (as in Naito et al., 2017).
- The protein levels and integrity of SorCS1 and SorCS1 Δ VPS10 should be assessed by Western blot to ensure that the inability of the latter to rescue amyloid-beta oligomers synaptic dysfunctions is not due to lowered protein stability or proteolytic truncations.
- In Fig 1D, similar amounts of SorCS1 are detected with the anti-his antibody in the unbound fraction whether NRX1 or NRX1 Δ HRD are used as bait. Using the anti-SorCS1 antibody, no signal is observed in the bound fraction of NRX1 Δ HRD. Can the authors explain this discrepancy? Why did they use a different antibody to analyze bound and unbound fractions?
- Fig 2A-D. In these experiments, the authors consider the monomeric concentration equivalent of A β oligomers; the concentration of particles is therefore much lower which may explain why >1000 nM of A β oligomers are needed to displace SorCS1 from NRX1beta. This issue should be mentioned and discussed.
- Fig 3C. This figure shows that SorCS1 and HA-NRX1beta are both expressed at the axon surface. To demonstrate proper colocalization, the authors should provide higher magnification images and quantifications (e.g. Pearson or Mander's correlation coefficients)
- Fig 6C-D. The decrease of PD95 and VGLUT1 puncta size in A β oligomers treated IRES-GFP neurons, and the rescue by SorCS1 transfection was not clear for me at first glance. I suggest the authors think of another way to represent these data that makes it easier for the reader to appreciate what is measured and what is happening.

Referee Cross-Comments:

Reviewer #3 raised a number of additional and important issues that I did not detect. I agree with this reviewer that these issues need to be addressed to support the findings and strengthen this manuscript.

Reviewer #3 (Comments to the Authors (Required)):

Lee and colleagues investigate the competitive interaction of the transmembrane protein SorCS1 and Amyloid- β -Oligomers (A β O) with β -Neurexins (Nrxn). SorCS1 is a neuronal sorting receptor previously shown to interact with Nrxn1 β and implicated in its intracellular targeting. SorCS1 has also been identified as an Alzheimer disease (AD) risk factor. A β accumulation is thought to underlie AD progression and synapses are vulnerable to A β . Nrxn are part of a synaptic organizer complex and interact with A β O. A β O induces Nrxn dysfunction and inhibits Nrxn dependent synaptic differentiation.

The authors demonstrate the previously shown Nrxn-SorCS1 interaction. They add new information by demonstrating that the interaction of SorCS1 with Nrxns is mediated by the Nrxn HRD domain. This domain conveys also interaction with A β O and using cellular assays, the authors demonstrate competitive binding of SorCS1 and A β O to Nrxn, axonal localization of SorCS1 and Nrxn and rescue of A β O-induced impairment of Nrxn dependent synapse formation by SorCS1.

The manuscript addresses an interesting issue, provides valuable information for understanding a possible function of SorCS1 in AD and suggests a protective role of SorCS1 by forming a synaptic complex with Nrxn. The manuscript is well written, applied methods are sound and employ mainly cell cultures and primary neuronal cultures.

However, there are a number of concerns, as there is a missing experimental control for the SorCS1 Δ VPS10 construct, questionable interpretation of some data and several images need to be improved to support the findings. Finally, the specification of the splice variant SorCS1b in parts the text would strengthen the manuscript.

Major points:

1. SorCS1 is expressed as different splice variants. These differ in their cytoplasmic domains. The here used SorCS1b seems to be the variant with highest cell surface expression whereas other variants localize more to endosomes. As already observed for Nrns, splicing adds additional functional complexity and it is sometimes difficult to distinguish splice variant specific effects as all variants of SorCS1 share the same extracellular domain.

The manuscript is sometimes vague if SorCS1, the overall protein including all splice variants, is described or specifically SorCS1b. This pertains e.g. the abstract in which SorCS1b is not mentioned at all. The authors should improve the text regarding this important issue.

2. Figure 1. D: Input Western blot: NRX1 β - Δ HRD-Fc runs higher than NRX1 β -Fc, although NRX1 β - Δ HRD-Fc lacks the HRD domain (amino acids 50-83 in NRX1 β). This is surprising as NRX1 β - Δ HRD-Fc lacks a domain and should have a reduced molecular weight. Please clarify.

3. Figure 2 E, G: Subcellular localizations of SorCS1b-myc and SorCS1b Δ VPS10-myc is difficult to estimate from the images. A merged picture is not shown, but the myc signal appears not close to the surface labeling when compared to the surface HA signal. Actually, the myc signal even seems to appear rather at the ER, what would suggest incorrect protein folding and would make the data obsolete. Images that are more convincing have to be selected.

The authors demonstrate surface localization of the SorCS1b-myc construct. Please compare Fig. EV3. In the SorCS1b Δ VPS10-myc a domain is deleted. Such a deletion can interfere with correct protein folding and may cause ER retention of the construct. Please demonstrate also for the SorCS1b Δ VPS10-myc its surface localization. Otherwise, experiments using this construct will be inconclusive. As the construct already exists, experiments can be performed in a few weeks.

4. Figure EV2: The selected image showing myc staining of SorCS1b-myc should be improved. The cells appear smaller than the ones shown for myc-CD4 and myc-NRX1 β and it is unclear if this is really surface expression.

5. Figure EV4: Title: NLGN1-Fc-coated beads recruit both HA-NRX1 β and SorCS1 on the axon surface.

From the figure legend, it is unclear that only parts of the axon are shown. The authors should present also an overview low magnification image of the cultured neurons as well as a MAP2 staining.

The authors use an anti-SorCS1 antibody for surface labeling of the SorCS1b-myc and SorCS1b Δ VPS10-myc constructs. Did they observe also surface staining in untransfected neurons?

The authors observe more surface staining in SorCS1b-myc expressing cells and present a figure in which two NLGN1-Fc-coated beads are located next to each other. In contrast, they observe in SorCS1b Δ VPS10-myc expressing cells less surface localization in the vicinity of only one NLGN1-Fc-coated bead. Importantly, in the SorCS1b Δ VPS10-myc expressing cells some surface staining can be detected. The observed difference could be due to a dose dependent effect, the difference of providing one or two coated beads. As these are representative images, I suggest showing additional ones. The authors may support their findings also by quantitative analyses of the fluorescent signal.

6. Figure 4A, C, E: The selected representative images do not fully support the conclusions.

E.g. 4A 2nd panel, as expected the GFP and the VGLUT1 signal are increased and structured in the area of the NLGN1-bead. 4A 4th panel the GFP signal is indistinguishable from the background signal produced by the bead, as it is similar to the one produced by the IRES-GFP vector alone (3rd panel), in contrast, the VGLUT1 signal is increased in a broader area (these could be additional non-transfected axons?).

I suggest to replace some of the images with more representative ones and to present as supplemental information larger parts of the cultures in lower magnification images.

7. Figure 6: Please show also a merged image of GFP and VGLUT1 to demonstrate that endogenous VGLUT1 signals match transfected axons.

Minor points:

1. On page 5, the authors conclude that the pull-down experiments "indicate a direct protein interaction". This is misleading as pull down experiments can also be successful through intermediate proteins found in a complex.

2. Several plasmids were obtained from Addgene and the respective scientist who originated the plasmid indicated, but not cited. In accordance with the Addgene guidelines, please cite the article in which the plasmid was initially described.

3. Traummüller, Gomez, Nguyen and Scheiffele also reported in 2016 the Nrnx SorCS1/2 interaction and should be cited in this context.

First, we thank all the reviewers for their careful reviewing with many positive comments and very helpful and constructive suggestions. We performed additional experiments to answer the reviewers' comments and to strengthen our conclusions. These new data have been integrated into our revised manuscript. Below are point-by-point answers with the reviewers' comments in *italic font* and the responses and descriptions of changes made in the manuscript in **bold font.**

Reviewer #1 (Comments to the Authors (Required)):

REVIEW - AK LEE ET AL

GENERAL COMMENTS

This is a top 10% piece of research. It concerns a topic that is significant and should be of wide interest, namely, the mechanism of action by which the synaptotoxicity of amyloid beta oligomers (A β O) is mediated. The issue is relevant to the cause of Alzheimer's disease and presents the possibility of a novel therapeutic target. Results reveal important molecular details regarding the toxic interaction of A β O with neurexin and shows the competitive cis action of SorCS1 in blocking this interaction. The data are of high quality and support the conclusion. The findings are supported by excellent writing, which clearly states the problem and its solution. The flow of the paper is logical, and each paragraph communicates its message effectively. The methods are described in detail. Overall, this manuscript is a model of matching conclusions with evidence, of good organization, and of clarity.

ESSENTIAL CHANGE - Fix the Scholarship

• Simple but significant emendation is needed before I can recommend this manuscript be accepted - the time and effort required are minimal. The manuscript concerns the mechanism of synapse damage caused by amyloid beta oligomers (A β O). A β O are neurotoxins widely considered to play a pathogenic role in Alzheimer's disease. The study is introduced in a scholarly manner. The authors credit numerous earlier works for establishing the foundation to the current investigation. Their scholarship is not adequate, however, and seminal work has been neglected, despite it being well known and highly germane to this study. These salient publications also were ignored in an excellent earlier study by this group, published in Scientific Reports. This gap in scholarship should be corrected before the manuscript is published.

• The lead paragraph covers the cogent milestones that define the field. The milestones are well-chosen and set the stage for the current work. They appear in the following order. (1) Accumulation of "toxic A β peptides" is a characteristic of AD ... major milestones are then reviewed, and the authors cite studies linking A β O to the following neural damage: (2) Synapse dysfunction and loss; (3) Decreases in pre-/postsynaptic proteins; (4) Decreases in synaptic vesicle recycling; (5) Decreases in synaptic spines; (6) Impaired LTP; (7) Enhanced LTD; (8) In vivo substantiation of A β O impact on synaptic protein expression, dendritic spine density, and

LTP.

- *Based on the cited evidence, the paragraph concludes "...synapses are vulnerable to A β , and understanding the molecular mechanism that underlie this vulnerability is crucial for explain how A β induces synapse pathology and how A β -induced synapse pathology could be ameliorated."*

- *In reviewing the literature, the authors have missed a set of benchmark studies that need to be added. These seminal studies are well known and germane to the milestones of the first paragraph. Point by point:*

(1) Accumulation of toxic peptides in AD ... The "toxic" A β peptides are A β O_s, and the seminal works regarding accumulation of A β O_s in AD are Kaye et al, Science, 2003, and Gong et al, PNAS, 2003. (2,3,5) Synapse dysfunction and loss and decreases in pre-/postsynaptic proteins. Missing seminal studies, which showed that A β O_s target synapses, cause loss of key synaptic proteins, and induce spine degeneration are Lacor et al, J. Neurosci., 2004, 2007. (6) Impaired LTP. The missing study, which also introduced the hypothesis that A β O_s cause the neural damage leading to AD, is Lambert et al, PNAS, 1998. (7) Enhanced LTD. The missing study is Wang et al, Brain Res, 2002.

- *Given their scholarly overview, the authors should cite papers in which the A β O hypothesis was introduced (Lambert et al, PNAS, 1998) and cogently reviewed (Cline et al, JAD, 2018). They also should include the influential (but somewhat controversial) paper on A β O-induced memory loss (Lesne et al, Nature, 2006).*

Response: I agree with the reviewer's comment that, in the original manuscript, we missed the citation of key seminal studies. After carefully checking the papers suggested by the reviewer, we added citations for all except the Nature paper (Lesne et al, Nature 2006) into the Introduction section of our manuscript. As the reviewer pointed out, the Nature paper (Lesne et. al, 2006) is controversial (Science. 2022 Jul 22;377(6604):358-363), with Nature investigating its data authenticity and advising caution when using results reported therein. Because of this, we have not included a citation for it. We believe that the revised manuscript now has more complete and fair scholarship citation.

MINOR POINTS

- *A graphic illustrating their mechanism would be appealing.*

Response: We agree that mechanistic diagrams are useful and appealing, but decided not to add a graphic illustration because such illustrations sometimes overestimate presented data and/or mislead readers to biased data interpretation.

- *It could be useful for the authors to comment on the work by Ohnishi et al, PNAS 2015, which attributed the presynaptic mechanism of A β O_s to their binding to the Alpha3-NaK ATPase.*

Response: To elaborate on the therapeutic potential of presynaptic targeting for A β synaptic pathology and neuronal cell death, we added a new discussion paragraph including comment on the work regarding Alpha3-NaK ATPase on Page 17-18 (Word file manuscript) in the Discussion section.

- *The method for preparing A β O_s cites their earlier paper (Naito et al) but it appears to be based on the method introduced by Stine et al, JBC, 2003, which they should cite.*

Response: We have added this citation to the Materials and Methods section as follows:

“A β (1-42) (Cat# A-1002-2, 1 mg, r-peptide) and biotin-tagged A β (1–42) (Cat# AS-23523-05, 0.5 mg, Anaspec) were used to make oligomeric forms, as we did previously (Naito Y et al, 2017) based on the method described in an earlier study (Stine WB, Jr. et al, 2003).”

• The authors (and the field) should avoid referring to "A β " and to "amyloid" when discussing the action of "A β Os." The lead paragraph, e.g., states "A β induces synaptic pathology." As the authors make clear, it is not "A β " that induces the pathology - it is A β oligomers (A β Os). The field's literature suffers from incorrect, interchangeable, indiscriminate use of amyloid, A β , and A β Os.

Response: We have carefully checked all instances of “amyloid”, “A β ”, and “A β Os” in the manuscript to ensure that the proper terminology is used consistently.

• It might be useful to comment on limits of therapeutic benefits derivable from targeting the SorCS1/neurexin mechanism. For example, would targeting this mechanism help protect against other effects listed in their introduction (e.g., pathological removal of post-synaptic proteins or dysfunction in synaptic plasticity)?

Response: We have added a new discussion into the last paragraph of the Discussion section on Page 17-18 (Word file manuscript) that describes how potential interplay between the Alpha3-NaK ATPase, NRXs, SorCS1 and TrkB pathways could help protect against A β O-induced neuronal cell death.

• A section on abbreviations would be helpful.

Response: We have not added a list of abbreviations to comply with the journal's manuscript preparation guidelines, which state that abbreviations should be defined in brackets after their first mention in the text and not in a list of abbreviations.

• Occasionally, sentences are too long and complex.

Response: We have revised the manuscript to split up excessively long and complex sentences.

Reviewer #2 (Comments to the Authors (Required)):

This very interesting and well-conducted study nicely demonstrates a protective role for SorCS1 with respect to amyloid-beta oligomers induced synaptic pathology. The paper is well written and the conclusions made by the authors are for the most part supported by the data, without over-interpretations.

However, some issues need to be addressed before the paper can be considered for publication:

- The authors should provide some quality control data (as they did in Naito et al., 2017) with respect to the amyloid-beta oligomers preparations they used in this study. Some of the experiments, such as the amyloid-beta/SorCS1 competition for NRX1beta binding (Fig 2) would be strengthened by including a non-oligomeric amyloid-beta "fresh" sample (as in Naito et al., 2017).

Response: As supplementary Fig S4 in the revised manuscript, we have added images of western blots probed with the 6E10 antibody as quality control data regarding A β oligomer preparation. Also, we revised the second paragraph of the Result section as follows:

“We have previously discovered that the β -NRX HRD is also responsible for interaction of β -NRX with A β Os but not A β monomers (Naito Y et al, 2017), suggesting the possibility that SorCS1 and A β Os compete for binding to β -NRXs since they share a binding domain on β -NRXs. To test this, we performed cell-based competitive protein binding assays using oligomerized samples of biotin-conjugated A β peptides (biotin-A β O) (Fig S4).”

We agree with the reviewer’s comment that including a non-oligomeric amyloid-beta "fresh" sample condition could strengthen our data. However, we have not used fresh samples in competitive binding assays or any neuron-based pathology experiments for the following reasons. Fresh samples always include a significant level of low molecular weight (LMW) oligomer forms, as shown in our previous paper (Naito Y., et al., Sci. Rep. 2017) and in the current quality control image (Fig S4), as well as in several previous key studies about amyloid- β oligomer (A β O) receptors such as PrP^c (Lauren J et al, Nature, 2009) and PirB (Lim T et al, Science 2013). This makes interpretation of the effects of A β monomers (non-oligomeric form) in competitive binding assays and neuron-based pathological experiments challenging if not impossible. In cell-surface binding assays using fresh samples (see below, unpublished data), we observed significant binding signals on COS-7 cells expressing HA-NRX1 β (approximately equivalent to 25-50% of the binding signal observed with A β Os). However, their binding does not appear to match that of typical saturable ligand-receptor binding in the nanomolar range, but rather is non-specific binding or of lower affinity than typical for ligand-receptor binding. Furthermore, in previous pull-down experiments using purified NRX1 β -Fc and A β samples containing HMW and LMW oligomers as well as monomers, we clearly demonstrated that NRX1 β -Fc does not bind to A β monomers (Naito Y., et al., Sci. Rep. 2017). Thus, A β monomers (non-oligomeric A β) would be unable to affect SorCS1-NRX1 β interaction. Finally, previous studies that addressed the roles of A β O receptors such as PrP^c and PirB also used vehicle or no A β O condition, but not fresh samples, as a negative control.

- The protein levels and integrity of SorCS1 and SorCS1 Δ VPS10 should be assessed by Western blot to ensure that the inability of the latter to rescue amyloid-beta oligomers synaptic dysfunctions is not due to lowered protein stability or proteolytic truncations.

Response: We first made efforts on Western blot using the lysates prepared from multiple coverslips of low-density cultured hippocampal neurons transfected with SorCS1b-IRES-GFP and SorCS1b Δ -IRES-GFP by AMAXA nucleofection, to match the experimental conditions used in the neuron-based A β O pathology experiments in this study. However, we were not able to detect any reliable SorCS1 band probably due to the very limited amount of protein in these samples. We therefore next performed immunocytochemical analysis of surface signals for SorCS1b and SorCS1b Δ VPS10 on axons of transfected neurons under the same experimental conditions. As shown in the new supplementary figure (Fig S9), like SorCS1b, SorCS1b Δ VPS10 can be targeted to the axon surface. Further, the quantitative data indicate that axonal expression of SorCS1b Δ VPS10 is comparable to that of SorCS1. These data suggest that the lack of rescue ability of SorCS1b Δ VPS10 is not due to insufficient axonal expression caused by lowered protein stability and/or proteolytic truncations. Further, our new colocalization data (Fig 3C and D) show that the colocalization level of SorCS1b Δ VPS10 with HA-NRX1 β on the axon surface is significantly lower than that of SorCS1b with HA-NRX1 β . Together with a previous study (Savas JN et al, Neuron, 2015), these new data support the conclusion that the deletion of VPS10 abolishes extracellular SorCS1-NRX1 β interaction without affecting surface expression of SorCS1b. Therefore, we believe that SorCS1b Δ VPS10-IRES-GFP is a good negative control in this study.

- In Fig 1D, similar amounts of SorCS1 are detected with the anti-his antibody in the unbound fraction whether NRX1 or NRX1 Δ HRD are used as bait. Using the anti-SorCS1 antibody, no signal is observed in the bound fraction of NRX1 Δ HRD. Can the authors explain this discrepancy? Why did they use a different antibody to analyze bound and unbound fractions?

Response: We carefully checked again what antibody we used and verified that we used anti-His antibody for the immunoblots for both bound SorCS1-His and unbound SorCS1-

His (See also Fig S2). We have corrected the figure labels and thank the reviewer for pointing out this typo in Fig 1D.

Regarding the amount of SorCS1 in the unbound fraction, we agree with the reviewer's point that we should observe more unbound SorCS1-His in the NRX1 β Δ HRD-Fc lane than in the NRX1 β -Fc lane. However, one possibility is that SorCS1-His may be present in excess relative to NRX1 β -Fc in the pull-down assays, and some small fraction of the total is pulled-down in the experiment, leaving similar appearing levels in the unbound fractions regardless of whether pull-down occurred. In addition, we reviewed the original raw anti-His immunoblot image and found that the contrast enhancement of the anti-His unbound image was excessive in Fig 1D of the original manuscript. We have therefore replaced the image with one that more closely matches the raw data. We have also added the raw full membrane images of both the anti-His blot and the anti-Fc blot in Fig S2 for data transparency.

As shown in the revised image in Fig 1D and the raw image in Fig S2, the unbound SorCS1 band intensity in all three conditions is weak. This is because those samples are supernatant fractions after pulling down protein complexes, and we used the supernatant fractions without concentrating the unbound SorCS1-His on His affinity beads. Therefore, the supernatant sample contains insufficient protein for reliable quantitative assessment. Instead, our main purpose in showing the unbound fraction image is to qualitatively demonstrate the presence of SorCS1-His proteins in all three conditions before the pull-down step.

- Fig 2A-D. In these experiments, the authors consider the monomeric concentration equivalent of Abeta oligomers; the concentration of particles is therefore much lower which may explain why >1000 nM of Abeta oligomers are needed to displace SorcS1 from NRX1beta. This issue should be mentioned and discussed.

Responses: According to our quality control data for A β oligomer preparation, the molecular weight (MW) of biotin-A β O is likely to be 120-260 kDa (with a median of roughly 150 kDa), suggesting that the biotin-A β O MW would be around 30 times more than biotin-A β monomers (around 5 kDa). Thus, as the reviewer pointed out, the minimum concentration of A β O for binding competition might be less than 100 nM (roughly estimated, it could be 66 nM; 2000 nM divided by 30, the MW factor). In the revised manuscript, we added the above consideration into the Results section (Page 6) as follows:

“(from 0 nM to 2,000 nM, monomer equivalent, which corresponds to 0 nM to 66 nM 150 kDa oligomers).”

- Fig 3C. This figure shows that SorCS1 and HA-NRX1beta are both expressed at the axon surface. To demonstrate proper colocalization, the authors should provide higher magnification images and quantifications (e.g. Pearson or Mander's correlation coefficients)

Response: We have added quantification of the colocalization experiments using Pearson correlation coefficients (PC). We found that the PC value of SorCS1b and HA-NRX1 β was more than 80%, indicating a high degree of colocalization. Further, the PC

values of the mutants were significantly lower than that of SorCS1b and HA-NRX1 β and comparable to the PC value between SorCS1b and GFP, which represents no interaction. These new quantitative data significantly strengthen our claim of SorCS1b-NRX1 β colocalization on the axon surface and also added new information that the colocalization depends on their extracellular interaction, as colocalization was negligible in the presence of NRX1 β HRD or SorCS1VPS10. We are very grateful to the reviewer for this comment that allowed us to obtain such very convincing and important evidence.

- Fig 6C-D. The decrease of PSD95 and VGLUT1 puncta size in A β oligomers treated IRES-GFP neurons, and the rescue by SorCS1 transfection was not clear for me at first glance. I suggest the authors think of another way to represent these data that makes it easier for the reader to appreciate what is measured and what is happening.

Response: Bar graphs might be easier to interpret. However, they would show the average and SEM of averaged data of VGLUT1 or PSD-95 puncta size. Averaging of the average values of sample data occludes sample variations per subject (in this case, the variation in puncta size in each dendritic segment), resulting in reduced statistical power and causing Type II statistical error. We have therefore retained the cumulative distribution curve plots. But, to make it a bit easier for readers to appreciate the data, we changed a line color to more clearly show no significant change in the cumulative distribution of VGLUT1 and PSD-95 puncta size between IRES-GFP with vehicle (black) and SorCS1-IRES-GFP with A β Os (light green).

Referee Cross-Comments:

Reviewer #3 raised a number of additional and important issues that I did not detect. I agree with this reviewer that these issues need to be addressed to support the findings and strengthen this manuscript.

Response: We addressed all of reviewer #3's comments as shown below.

Reviewer #3 (Comments to the Authors (Required)):

Lee and colleagues investigate the competitive interaction of the transmembrane protein SorCS1 and Amyloid- β -Oligomers (A β O) with β -Neurexins (Nrxn). SorCS1 is a neuronal sorting receptor previously shown to interact with Nrxn1 β and implicated in its intracellular targeting. SorCS1 has also been identified as an Alzheimer disease (AD) risk factor. A β accumulation is thought to underlie AD progression and synapses are vulnerable to A β . Nrxn are part of a synaptic organizer complex and interact with A β O. A β O induces Nrxn dysfunction and inhibits Nrxn dependent synaptic differentiation.

The authors demonstrate the previously shown Nrxn-SorCS1 interaction. They add new information by demonstrating that the interaction of SorCS1 with Nrxns is mediated by the Nrxn HRD domain. This domain conveys also interaction with A β O and using cellular assays, the authors demonstrate competitive binding of SorCS1 and A β O to Nrxn, axonal localization of SorCS1 and Nrxn and rescue of A β O-induced impairment of Nrxn dependent synapse formation by SorCS1.

The manuscript addresses an interesting issue, provides valuable information for understanding a possible function of SorCS1 in AD and suggests a protective role of SorCS1 by forming a synaptic complex with Nrxn. The manuscript is well written, applied methods are sound and employ mainly cell cultures and primary neuronal cultures.

However, there are a number of concerns, as there is a missing experimental control for the SorCS1b Δ VPS10 construct, questionable interpretation of some data and several images need to be improved to support the findings. Finally, the specification of the splice variant SorCS1b in parts the text would strengthen the manuscript.

Major points:

1. SorCS1 is expressed as different splice variants. These differ in their cytoplasmic domains. The here used SorCS1b seems to be the variant with highest cell surface expression whereas other variants localize more to endosomes. As already observed for Nrxns, splicing adds additional functional complexity, and it is sometimes difficult to distinguish splice variant specific effects as all variants of SorCS1 share the same extracellular domain. The manuscript is sometimes vague if SorCS1, the overall protein including all splice variants, is described or specifically SorCS1b. This pertains e.g. the abstract in which SorCS1b is not mentioned at all. The authors should improve the text regarding this important issue.

Response: We have revised the manuscript to use more precise terminology when referring to the various forms of SorCS1. When describing data using soluble SorCS1 ectodomain, we use “SorCS1 ectodomain”, which is shared by all isoforms of SorCS1. For other data using portions other than the common domain, we have made sure to specify that “SorCS1b” constructs were used, including in the abstract.

2. Figure 1. D: Input Western blot: NRX1 β - Δ HRD-Fc runs higher than NRX1 β -Fc, although NRX1 β - Δ HRD-Fc lacks the HRD domain (amino acids 50-83 in NRX1 β). This is surprising as NRX1 β - Δ HRD-Fc lacks a domain and should have a reduced molecular weight. Please clarify.

Response: We were also surprised by this apparent discrepancy in molecular weight. We performed three independent Fc immunoblot experiments using three independent batches of NRX1 β Δ HRD-Fc and NRX1 β -Fc. First, we found that in all experiments

including the ones for which the data is shown in Fig 1D, the NRX1 β Δ HRD-Fc band appeared a bit higher on the blot than the NRX1 β -Fc band (Fig S3A, B), showing that this migration pattern is consistent between batches. Then, we performed another western blot using an antibody that recognizes the HRD of NRX1 β (Neuromab; clone N170A/1) and found that this antibody revealed a band in the NRX1 β -Fc sample lane, but not in the NRX1 β Δ HRD-Fc lane, confirming that the NRX1 β Δ HRD-Fc preparation is indeed lacking the HRD. In addition, through cell surface binding assays, we confirmed that the prepared NRX1 β Δ HRD-Fc protein binds to neuroligin1/2 (Fig S3C). Together, these data strongly indicate that the protein preparations are the correct proteins. We have added the above validation data as supplementary Fig S3 and added some descriptions into the Materials and Methods section on page 23-24 (Word file manuscript).

Next, we used the Protein Molecular Weight (https://www.bioinformatics.org/sms/prot_mw.html) tool to predict the molecular weight (MW) of the NRX1 β HRD (amino acid (AA) sequence: LGAHHHHFGSSKHHSVPVPIAIYRSPASLRGGHA) and found that it is only 3.67kDa. Such a small difference would be difficult to detect in a protein that runs between 50-75 kDa, even if the migration of proteins on SDS-PAGE depended solely on MW. And, according to previous studies, migration on SDS-PAGE does not always correlate with MW. This phenomenon is called “gel shifting” (Rath A et al, Proc Natl Acad Sci 2009), and is common for histidine-rich proteins (Shelake RM et al, PLoS One 2017). However, as histidine is a positively charged amino acid residue, the gel shifting of histidine-rich proteins usually happens as anomalous “slower” migration, and that is opposite to the phenomenon we observed here (NRX1 β -Fc has more histidine but migrates faster than NRX1 β Δ HRD-Fc). We cannot find any other explanation for these results, but, the unexpectedly slow migration of the mutant protein suggests that the NRX1 β HRD may have very unique molecular properties, which would be crucial for better understanding NRX1 β HRD-based protein interactions in future studies.

3. Figure 2 E, G: Subcellular localizations of SorCS1b-myc and SorCS1b Δ VPS10-myc is difficult to estimate from the images. A merged picture is not shown, but the myc signal appears not close to the surface labeling when compared to the surface HA signal. Actually, the myc signal even seems to appear rather at the ER, what would suggest incorrect protein folding and would make the data obsolete. Images that are more convincing have to be selected. The authors demonstrate surface localization of the SorCS1b-myc construct. Please compare Fig. EV3. In the SorCS1b Δ VPS10-myc a domain is deleted. Such a deletion can interfere with correct protein folding and may cause ER retention of the construct. Please demonstrate also for the SorCS1b Δ VPS10-myc its surface localization. Otherwise, experiments using this construct will be inconclusive. As the construct already exists, experiments can be performed in a few weeks.

Response: By performing surface SorCS1 immunostaining, we confirmed that SorCS1b Δ VPS10-myc can be also targeted to the COS-7 cell membrane (Supplementary Fig S4C) and the neuron surface (Fig. 3C). Also, in Fig 2E and G, we have replaced the images for SorCS1b Δ VPS10-myc with better ones.

4. Figure EV2: The selected image showing myc staining of SorCS1b-myc should be improved. The cells appear smaller than the ones shown for myc-CD4 and myc-NRX1 β and it is unclear if this is really surface expression.

Response: In the revised supplementary Fig S6, we selected a better image for SorCS1b-myc with surface-like signals and also reselected ones for myc-CD4 and myc-NRX1 β . Further, we updated the quantitative results to include data from two independent experiments.

Compared to typical type I transmembrane proteins, surface trafficking of SorCS1 is likely to be much less efficient even though we confirmed SorCS1b expression on the COS cell surface by immunostaining using SorCS1 antibody without cell permeabilization (Fig S5). Further, previous studies have shown that SorCS1 is efficiently shed by metalloproteases and γ -secretases (Hermey G et al, *Biochem J* 2006; Nyborg AC et al, *Mol Neurodegener*, 2006; Willnow TE et al, *Nat Rev Neurosci*, 2008). Considering these properties of SorCS1 and the reviewer's comment, we think that our previous conclusion that SorCS1 does not bind to A β O might be an overstatement of our currently limited data because it remains possible that the surface expression level of SorCS1b-myc may be insufficient for definitive detection of A β O binding. Therefore, we revised the Result section on Page 7 and the title of the Fig S6 legend to be more descriptive manner.

5. *Figure EV4: Title: NLGN1-Fc-coated beads recruit both HA-NRX1 β and SorCS1 on the axon surface.*

From the figure legend, it is unclear that only parts of the axon are shown. The authors should present also an overview low magnification image of the cultured neurons as well as a MAP2 staining.

The authors use an anti-SorCS1 antibody for surface labeling of the SorCS1b-myc and SorCS1b Δ VPS10-myc constructs. Did they observe also surface staining in untransfected neurons?

The authors observe more surface staining in SorCS1b-myc expressing cells and present a figure in which two NLGN1-Fc-coated beads are located next to each other. In contrast, they observe in SorCS1b Δ VPS10-myc expressing cells less surface localization in the vicinity of only one NLGN1-Fc-coated bead. Importantly, in the SorCS1b Δ VPS10-myc expressing cells some surface staining can be detected. The observed difference could be due to a dose dependent effect, the difference of providing one or two coated beads. As these are representative images, I suggest showing additional ones. The authors may support their findings also by quantitative analyses of the fluorescent signal.

Response: To show that NLGN1-Fc-coated beads recruit SorCS1b through an extracellular SorCS1b-NRX1 β interaction, we added new experimental data with fluorescent intensity quantification by performing protein clustering assays (Fig 2I-K). In these assays, we used SorCS1b-IRES-GFP constructs, rather than myc-tagged constructs, and immuno-stained the samples for MAP2 as well as surface SorCS1 and surface HA-NRX1 β . We found that NLGN1-Fc-coated beads, but not Fc-coated beads (a negative control), induced the clustering of both HA-NRX1 β and HA-NRX1 β Δ HRD. These results are consistent with those of a previous structural study showing that the HRD is supposed to be dispensable for NLGN1-NRX1 β interaction (Rudenko G et al, *Cell*, 1999). In addition, we found that NLGN1-Fc beads induced co-clustering of SorCS1b with HA-NRX1 β , but not of SorCS1b with HA-NRX1 β Δ HRD or of SorCS1b Δ VPS10 with HA-NRX1 β . These new data suggest that a cis-complex of SorCS1b and HA-NRX1 β on the axon surface can make a complex with NLGN1 through HA-NRX1 β . The revised figures (Fig 2I-K in the revised manuscript) show lower magnification images with MAP2 staining as

well as GFP signal to distinguish between axons and dendrites better than in Figure EV4 of the original manuscript, and each image shows three beads.

In the case of untransfected hippocampal neurons, we were not able to detect significant reliable immunoreactivity for surface SorCS1 (see also Fig 3A and B). When we increased the exposure time or the gain during imaging, some weak signal appeared. However, as we don't have SorCS1 knockout mice/neurons or any other SorCS1 knockdown tool, we are not sure if these signals are really endogenous SorCS1 expressed on the surface or just non-specific signals. Due to this technical limitation, the present study has focused on gain-of-function (GOF) approaches to investigate the rescue effects of SorCS1 on A β pathology. This also has the further benefit of being more applicable to *in vivo* rescue strategies in future studies. For the manuscript revision, we have added a description about the results of neurons transfected with IRES-GFP on Page 9 as follows:

“As a negative control, there was no apparent surface SorCS1 signal in neurons transfected with the empty IRES-GFP vector (Fig 3A and B), indicating that the surface SorCS1 signals observed in neurons transfected with SorCS1b-IRES-GFP are due to exogenous SorCS1b expression rather than non-specific signals or endogenous SorCS1 expression.”.

6. Figure 4A, C, E: The selected representative images do not fully support the conclusions. E.g. 4A 2nd panel, as expected the GFP and the VGLUT1 signal are increased and structured in the area of the NLGN1-bead. 4A 4th panel the GFP signal is indistinguishable from the background signal produced by the bead, as it is similar to the one produced by the IRES-GFP vector alone (3rd panel), in contrast, the VGLUT1 signal is increased in a broader area (these could be additional non-transfected axons?).

I suggest to replace some of the images with more representative ones and to present as supplemental information larger parts of the cultures in lower magnification images.

Response: We replaced the following images with better ones, as the reviewer advised.

- Figure 4A: NLGN1-Fc_vehicle_IRES-GFP (second column), NLGN1-Fc_A β O_SorCS1b-IRES-GFP (fourth column).
- Figure 4C: NLGN1-Fc_A β O_SorCS1b-IRES-GFP (fourth column).
- Figure 4E: Fc_vehicle_IRES-GFP (first column), Slitrk2-Fc_vehicle_IRES-GFP (second column), Slitrk2-Fc_vehicle_IRES-GFP (second column)

Further, we added supplementary Fig S7 to show lower magnification images of artificial synapse formation assays using NLGN1-Fc-coated beads with immunostaining for MAP2 as well as VGLUT1.

7. Figure 6: Please show also a merged image of GFP and VGLUT1 to demonstrate that endogenous VGLUT1 signals match transfected axons.

Response: We added the merged images of GFP (magenta) and VGLUT1 (green) into Fig 6A, which allow us to confirm that the majority of the VGLUT1 puncta signal overlaps with transfected axons.

Minor points:

1. On page 5, the authors conclude that the pull-down experiments "indicate a direct protein interaction". This is misleading as pull down experiments can also be successful through intermediate proteins found in a complex.

Response: We agree that we cannot exclude the possibility that intermediate proteins may be still present even after Fc protein concentration by Protein-G Dynabeads. Therefore, we revised our conclusion as shown below.

"These results provide further support for a protein interaction between the SorCS1 ectodomain and the NRX1 β ectodomain through its HRD, consistent with the results of the cell surface binding assays."

2. Several plasmids were obtained from Addgene and the respective scientist who originated the plasmid indicated, but not cited. In accordance with the Addgene guidelines, please cite the article in which the plasmid was initially described.

Response: We added the citation of the original paper for Addgene plasmids

- pCAG-GFP (kindly provided by Dr. Connie Cepko through Addgene (Matsuda T & Cepko CL, 2007))
- NRX1 β S4(-)-Fc from Dr. Peter Scheiffele (University of Basel) via Addgene (Scheiffele P et al, 2000)

3. Traunmüller, Gomez, Nguyen and Scheiffele also reported in 2016 the Nrnx SorCS1/2 interaction and should be cited in this context.

Response: We added the Science paper (Traunmuller L et al, 2016) into the Introduction and revised the sentence as shown below.

"Previously isolated in proteomics studies as an NRX1 β binding protein (Savas JN et al, 2015, Traunmuller L et al, 2016), SorCS1 interacts with NRX1 β through the SorCS1 VPS10 domain to promote surface expression of NRX1 β on axons (Savas JN et al, 2015)."

Other changes:

We added the Summary blurb in the manuscript text as shown below.

"The protein sorting receptor SorCS1 shields the synapse organizer β -neurexins from amyloid- β oligomers (A β O) to alleviate A β O-induced synaptic pathology."

In the Acknowledgement section, we added the acknowledgement of Nicolas Chofflet for his technical supports instead of listing his name in the author list upon his request with the consideration of the CRediT. We also added the information about the scholarships to N.Y. and H.K.

January 11, 2023

RE: Life Science Alliance Manuscript #LSA-2022-01681-TR

Dr. Hideto Takahashi
Montreal Clinical Research Institute
110 avenue des Pins Ouest
Montreal, Quebec H2W 1R7
Canada

Dear Dr. Takahashi,

Thank you for submitting your revised manuscript entitled "SorCS1 inhibits amyloid- β binding to neurexin and rescues amyloid- β -induced synaptic pathology". We would be happy to publish your paper in Life Science Alliance pending any final revisions necessary to meet our formatting guidelines.

A. FINAL FILES:

B. MANUSCRIPT ORGANIZATION AND FORMATTING:

Sincerely,

Reviewer #2 (Comments to the Authors (Required)):

I would like to commend the authors for their efforts in addressing my and the other reviewer's concerns. They have satisfactorily addressed the issues I raised and I found the manuscript very much improved and acceptable for publication.

Reviewer #3 (Comments to the Authors (Required)):

The authors have addressed all concerns I raised and to me also all points raised by the other reviewers. The authors amended the manuscript significantly and convincingly. I thank the authors for this thorough revision and wholeheartedly support the publication of this manuscript.

January 16, 2023

RE: Life Science Alliance Manuscript #LSA-2022-01681-TRR

Dr. Hideto Takahashi
Montreal Clinical Research Institute
110 avenue des Pins Ouest
Montreal, Quebec H2W 1R7
Canada

Dear Dr. Takahashi,

Thank you for submitting your Research Article entitled "SorCS1 inhibits amyloid- β binding to neurexin and rescues amyloid- β -induced synaptic pathology". It is a pleasure to let you know that your manuscript is now accepted for publication in Life Science Alliance. Congratulations on this interesting work.

DISTRIBUTION OF MATERIALS:

Again, congratulations on a very nice paper. I hope you found the review process to be constructive and are pleased with how the manuscript was handled editorially. We look forward to future exciting submissions from your lab.

Sincerely,
